

# The role of geochemistry in organic carbon stabilization in tropical rainforest soils

Mario Reichenbach[1], Peter Fiener[1], Gina Garland[2], Marco Griepentrog[2], Johan Six[2], and Sebastian Doetterl[1,2]

[1]Institute of Geography, Augsburg University, Augsburg, 86159 Germany

[2]Department of Environmental System Science, ETH Zurich, Zurich, 8092, Switzerland

*Correspondence to: S. Doetterl (sdoetterl@usys.ethz.ch)*

**Abstract**

Stabilization of organic carbon in soils (SOC) depends on several soil properties, including the soil weathering stage and the mineralogy of parent material. As such, tropical SOC stabilization mechanisms likely differ from those in temperate soils due to contrasting soil development. To better understand these mechanisms, we investigated SOC dynamics at three soil depths under pristine tropical african mountain forest along a geochemical gradient from mafic to felsic and a topographic gradient covering plateau, slope and valley positions. To do so we

conducted a series of soil C fractionation experiments in combination with an analysis of the geochemical composition of soil and a sequential extraction of pedogenic oxides. Relationships between our target and predicting variables were investigated using a combination of regression analyses and dimension reduction. Here, we show that reactive secondary mineral phases drive SOC properties and stabilization mechanisms together with, and sometimes more strongly than, other mechanisms such as aggregation or C stabilization by clay content. Key

mineral stabilization mechanisms for SOC were strongly related to soil geochemistry, differing across the study regions. These findings were independent of topography in the absence of detectable erosion processes. Instead, fluvial dynamics and changed hydrological conditions had a secondary control on SOC dynamics in valley positions, leading to higher SOC stocks there than at the non-valley positions. We also detected fossil organic carbon (FOC) at several sites, constituting up to 52.0 ± 13.2 % of total SOC stock in the C depleted subsoil.

Interestingly, total SOC stocks for these soils did not exceed those of sites without FOC. Additionally, FOC decreased strongly towards more shallow soil depths, indicating decomposability of FOC by microbial communities under more fertile conditions. Regression analysis showed that variables affiliated with soil weathering, parent material geochemistry and soil fertility, together with soil depth, explained up to 75 % of the variability of SOC stocks and $\Delta^{14}$C. Furthermore the same variables explain 44 % of the variability in the relative

abundance of C associated with microaggregates versus free silt and clay associated C fractions. However, geochemical variables gained or retained importance for explaining SOC target variables when controlling for soil depth. We conclude that despite long-lasting weathering, geochemical properties of soil parent material leave a footprint in tropical soils that affects SOC stocks and mineral related C stabilization mechanisms. While identified stabilization mechanisms and controls are similar to less weathered soils in other climate zones, their

relative importance is markedly different in the investigated tropical soils.



## 1 Introduction

### 1.1 SOC research in the tropics

The tropics are considered potential tipping points for the climate-carbon (C) feedback due to their substantial C storage in the biosphere, fast C turnover and the associated potential C losses to the atmosphere. Despite this key relevance in the terrestrial C cycle and climate regulation, the tropics remain highly understudied (Schimel et al., 2015). This is especially true for tropical soils, which are estimated to contain approximately one third of global

soil organic carbon (SOC) (Köchy et al., 2015). Many interacting soil processes, both in temperate and tropical soils, are not adequately represented in C turnover models, such as the effect of soil aggregation on soil biota and SOC dynamics (van Keulen, 2001; Wood et al., 2012; Vereecken et al., 2016). Studies analyzing the effect of soil geochemistry on SOC dynamics and stabilization are also rare (Wattel-Koekkoek et al., 2003; Denef and Six, 2005; Zotarelli et al., 2005; Quesada et al., 2020) and such effects are not included in large-scale C cycle modelling

approaches (Vereecken et al., 2016). Most of these geochemical effect studies focus on mid latitudes in the northern hemisphere, while the specific conditions under tropical conditions with highly weathered soils remain relatively unknown (Schimel et al., 2015) and can differ greatly compared to temperate soils (Denef and Six, 2006; Denef et al., 2007). Thus, findings from mid latitudes are not easily transferable to tropical soils, since the potential in stabilizing SOC depends on geochemical soil properties that differ fundamentally between geo-

climatic zones as a function of pedogenesis. The lack of mechanistic understanding regarding SOC dynamics and their controlling factors creates substantial uncertainties when predicting the future of SOC stocks in the tropics (Schmidt et al., 2011; Shi et al., 2020).

### 1.2 Environmental and geochemical controls on SOC dynamics in tropical forests

SOC dynamics in tropical rainforests are characterized by high C input and fast C turnover rates (Pan et al., 2011; Wang et al., 2018). Carbon input to soils is mainly driven by root growth and litter production (Raich et al., 2006) both of which are often driven by climatic and hydrological variables that govern vegetation dynamics. Climatic factors through controlling soil temperature and moisture conditions can also greatly influence soil microbial activity and hence C mineralization and turnover (Davidson and Janssens, 2006; Zhang et al., 2011; Feng et al.,

2017). The accessibility of C for mineralization, however, is predominantly driven by several interacting mechanisms that can stabilize C in soils. For example, certain C compounds such as pyrogenic or aromatic C are recalcitrant against decomposition due their complex molecular structure and can remain stable for decades or even centuries (Mikutta et al., 2006; Knicker, 2011; Singh et al., 2012). Another C fraction that is considered chemically recalcitrant is fossil organic carbon (FOC), deposited during the formation of sedimentary rocks and

often hard to decompose (van der Voort et al., 2019; Kalks et al., 2020). Carbon can also be protected physically against decomposition by encapsulation within soil aggregates. Minerals can also increase the energetic barrier for microorganisms to overcome by forming organo-mineral associations (Oades, 1984; Oades, 1988; von Lutzow et al., 2007; Lehmann et al., 2007; Cotrufo et al., 2013). In particular, it has been shown that the availability of reactive mineral surfaces influences the formation of organo-mineral associations as well (Eusterhues et al., 2003;

Jagadamma et al., 2014; Angst et al., 2018). Furthermore, reactive and adsorptive surfaces not only contribute to chemical C stabilization but also favor the formation of soil aggregates (Simpson et al., 2004; Six et al., 2004; Chenu and Plante, 2006; Lehmann et al., 2007).



While these general types of stabilization in the tropics are similar to those in temperate soils, their relative
importance and abundance differs greatly due to contrasting weathering history (Six et al., 2002; Denef et al.,
2004). Most temperate soils have developed from young (peri)glacial sediments and relatively unweathered
bedrock (~15,000 years old). Tropical soils have often been exposed to chemical weathering for millions of years
if landforms are stable (Porder et al., 2005; Finke and Hutson, 2008). The resulting soil geochemistry in the tropics
is therefore often composed of end members of weathering products such as secondary minerals (i.e. 1:1 low
activity clays, kaolinite) and highly crystalline, pedogenic oxides (West and Dumbleton, 1970). Clay-sized
mineral fractions in tropical soils are composed of ~50 % pedogenic oxides, which is usually much higher than in
temperate soils (Ito and Wagai, 2017). While some studies in tropical regions have shown that variation in clay
content explains SOC stocks in kaolinitic soils (Quesada et al., 2020), others have shown that SOC stabilization
is not affected by clay quantity, but instead by the clay mineralogy (Bruun et al., 2010). The most important
identified stabilization mechanisms in kaolinitic tropical soils are mineral-organic associations with short range
ordered (SRO) pedogenic oxides (Kleber et al., 2005; Bruun et al., 2010; Martinez and Souza, 2019) which
stabilize 47 % to 63 % of the bulk SOC stocks in tropical forests (Kramer and Chadwick, 2018). Hence, differences
in mineralogy affect a number of key soil fertility parameters and also the way C is stabilized onto minerals,
ultimately impacting the interplay between mineral reactivity, microbial community structures and nutrient
dynamics (Six et al., 2002; Denef and Six, 2005; Doetterl et al., 2018). For example, SOC stabilization by mineral-
organic complexes in tropical soils are highly efficient as they appear in parallel and within highly stable soil
aggregates and pseudosand structures (Martinez and Souza, 2019; Quesada et al., 2020). Furthermore, kaolinitic
soils can form aggregates rapidly independent from biological processes due to electrostatic interactions between
1:1 clay minerals and oxides. But biological processes can lead to stronger organic bonds in soils with 1:2 clays,
promoting long-term stability (Denef and Six, 2005). This finding relates to the observation that SOC stabilized
in kaolinitic 1:1 clay soils turns over faster compared to SOC associated with 1:2 clay soils (Wattel-Koekkoek et
al., 2003). Hence, interactions between geochemistry, aggregation and mineral surface, governed by soil
weathering, need to be considered more prominently to understand SOC dynamics in temperate vs. tropical soils.

### 1.3 Topographic controls on SOC dynamics in tropical forests

In addition to larger-scale biogeochemical and climatic controls of C dynamics, in undulating landscapes soil
redistribution processes can highly influence SOC dynamics (van Hemelryck et al., 2010; Doetterl et al., 2016;
Wilken et al., 2017). Excessive erosion of topsoils on hillslopes often results in exposure of subsoils with low C
contents, which on the one hand can lead to dynamic C replacement (Harden et al., 1999) and on the other hand
may stimulate the decomposition of older SOC due to priming with fresh C inputs (Fontaine et al., 2004; Keiluweit
et al., 2015). Thereby, removal of weathered topsoils brings new mineral surfaces in contact with fresh C input
which could favor C sequestration due to organo-mineral associations (Doetterl et al., 2016), especially in highly
weathered tropical landscapes (Vitousek et al., 2003; Porder et al., 2005). These processes are of potentially great
importance for tropical soil systems, as an erosional rejuvenation of land surfaces can bring an entirely different
soil mineral composition in touch with the biological C cycle, and provide a geochemically entirely different
environment for C stabilization. Similarly, geogenic carbon that is brought to the surface might become
increasingly decomposed when brought in contact with more active microbial communities. Parallel to these
processes of soil denudation, at depositional sites in valleys and at footslopes, former topsoil SOC can become



buried by colluvial and alluvial sediments, potentially greatly decreasing microbial decomposition. However, the
fate of buried SOC depends greatly on the prevailing environmental conditions in the depositional sites and the
sedimentation rates (Gregorich et al., 1998; Berhe et al., 2007; Berhe et al., 2012). Topography can control
hydrological patterns in tropical rainforests (Silver et al., 1999; Detto et al., 2013). For example, high water tables
lead to lower soil oxygen levels in valley positions which in turn reduce microbial C decomposition and potentially
result in the accumulation of labile SOC. Furthermore, changes in soil water content can cause reductive
dissolution of iron oxides which ultimately affects organo-mineral associations (Berhe et al., 2012). Thus, the
interplay between environmental, geochemical and topographic conditions set the stage for C stabilization and
will most likely differ from temperate to tropical soils.

### 1.4 Study aims

In summary, our current understanding on how geochemistry and topography in highly weathered tropical soils
affects SOC stocks and stabilization mechanisms is still limited. This study thus aimed to better understand the
influence of topography and geochemical properties of soils developed from different parent materials on (i) SOC
stocks, (ii) SOC fractions and (iii) SOC stabilization mechanisms in tropical forest soil systems. In addition, (iv)
we assessed the contribution of FOC to SOC stocks in sedimentary rock derived soils using $\Delta^{14}C$. Within this
context, the following hypotheses were tested:

(i) Topography governs the patterns of geochemical soil properties and SOC stocks due to its effect on lateral
water and matter fluxes. Sloping positions will show lower SOC stocks compared to plateau positions as a result
of erosion. In contrast, valley positions will show higher SOC stocks compared to plateau positions due to
deposition of C-rich topsoil material and limited composition of SOC.

(ii) Chemical variability in parent material will result in contrasting geochemical soil properties that affect the
formation of C stabilization mechanisms (organo-mineral associations and soil aggregates), and hence govern
patterns of SOC stocks and stability. More SOC will be stabilized in geochemical regions where soils are prone
to the development of higher content of clay and pedogenic oxides relevant for the stabilization of C with minerals.
Despite deep and long lasting tropical weathering, we expect differences in soil chemistry that relate back to parent
material chemistry will have a significant impact on C stabilization in soil.

(iii) SOC stocks in soils developed from deeply weathered FOC bearing parent material will show high contents
of FOC in subsoils, but decreasing content in topsoils, as the decomposition of FOC is facilitated through priming
at the soil surface in the presence of more readily available nutrient sources and higher microbial activity.

## 2 Materials and Methods

### 2.1 Study region

The study region is located in the eastern part of the Congo basin and the western part of the blue Nile basin with
study sites located along the East African Rift Mountain System. Vegetation at all sites is dominated by primary
tropical mountain forests (Fig. 1b). Climate of the region is characterized as humid tropical (Koeppen Af-Am)
with a short dry season (i.e. only two months per year with < 50 mm precipitation). The tectonically active rift



system resulted in geochemically diverse parent material and in a heterogeneous hilly landscape.The study area

consists of parent material ranging from mafic to felsic magmatic bedrock as well as sedimentary rocks of mixed geochemical composition. The undulating landscape resulted in a variety of hydrological conditions at plateau, slope and valley positions. In combination, this makes the study area ideal to analyze the effect of soil geochemistry and topography on SOC stabilization mechanisms and stocks in  a variety of tropical soils.

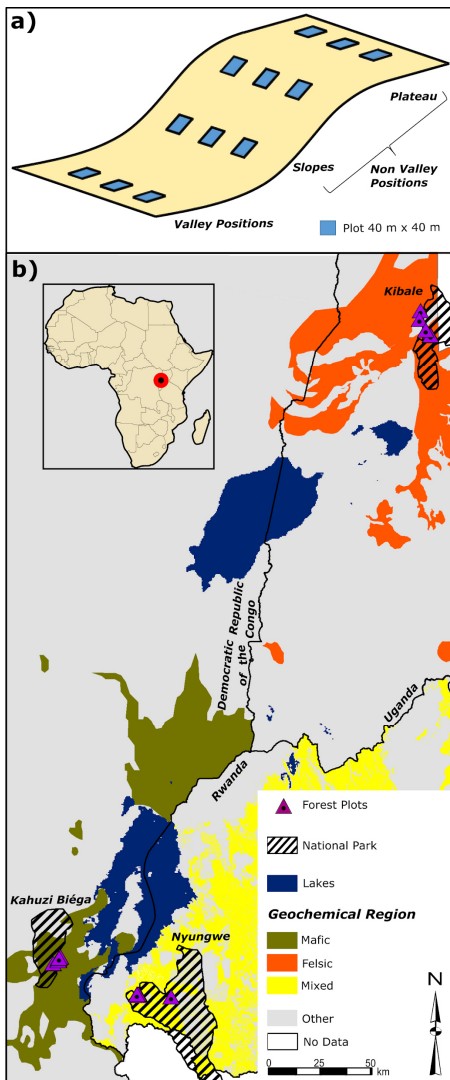

**Figure 1: (a) Study design used in each geochemical region. (b) Overview of study area with respect to soil parent material geochemistry (modified after Doetterl et al., 2021b).**

*Kahuzi-Biega*

The study region consisted of three main sites. The Kahuzi-Biega site (further called mafic site) is located in the

South Kivu Province of the Democratic Republic of the Congo (DRC) (-2.31439°S; 28.75246°E) with an altitude of 2220 ± 38 m.a.s.l and slopes ranging between 1-60 %. The parent material consists of mafic alkali-basalts



ranging with an age between 9-13 Ma (Schlüter and Trauth, 2006). According to FAO soil classification (FAO, 2014), typical soils in this region are ferralic nitisols and geric ferralsols. Vegetation is described as a closed broad-leaved semi-deciduous mountain forest (Verhegghen et al., 2012). Mean annual precipitation (MAP) is 1924 mm y$^{-1}$ and the mean annual temperature (MAT) is 15.3 °C (Fick and Hijmans, 2017).

*Nyungwe*

The Nyungwe site (further called mixed sedimentary site) is situated in the southwestern part of Rwanda (-2.463088°S; 29.103834°E) at 1909 ± 22 m.a.s.l. and with slopes ranging between 1-60 %. The parent material consists of mixed sedimentary rocks showing alternating layers of quartz-rich sandstone, siltstone and dark clay shists with an age between 1000-1600 Ma (Schlüter and Trauth, 2006). A specific feature of the sedimentary site is the presence of FOC in the parent material of soils ranging between 1.29-4.03 % C. FOC in these sediments is further characterized by a high C / N ratio (153.9 ± 68.5), depleted in N and free of $^{14}$C (due to the high age of sedimentary rock formation). Typical soils are geric ferralsols and fluvic gleysols. Vegetation is classified as an afromontane rainforest (van Breugel et al., 2020). MAP is 1702 mm y$^{-1}$ and MAT is 16.7 °C (Fick and Hijmans, 2017).

*Kibale*

The Kibale site (further called felsic site) is located in western Uganda (0.46225°N; 30.37403°E) with an altitude of 1324 ± 60 m.a.s.l and slopes between 3-55 %. The parent material consists of felsic gneissic granites with an age between 1600-2500 Ma (Schlüter and Trauth, 2006). Dominant soils are geric ferralsols. Vegetation is classified as Lake Victoria drier peripheral semi-evergreen Guineo-Congolian rainforest (van Breugel et al., 2020). The MAP is 1697 mm y$^{-1}$ and the MAT is 19.2 °C (Fick and Hijmans, 2017).

**2.2 Study design and soil sampling**

In the framework of project TropSOC (Doetterl et al., 2021a;b), soil sampling took place from March to June 2018, applying a stratified random sampling design with triplicate plots of 40 x 40 m across three topographic positions (i.e. plateau, slope, and valley, Fig. 1a) in each geochemical region. Note that because hillslopes were much larger landscape features, we sampled at both topslope and midslope positions. Slope steepness was measured at the center of each plot using a clinometer. Slope length at each plot was derived from a shuttle radar topography mission digital elevation model (SRTM-DEM) (NASA JPL, 2013) with a 30 m x 30 m resolution using the flow direction and flow length tool in ArcMap 10.6.1 (ESRI, USA). Slope length in Kahuzi-Biega ranged from 29 to 127 m, in Nyungwe from 72 to 338 m and in Kibale between 29 and 373 m. No evidence of soil erosion could be observed during the field survey within the plots and all soil samples were free of carbonates and inorganic C. Soils were described for every topographic position per geochemical region following WRB classification (FAO, 2014). To describe the chemical composition of the parent material, unweathered bedrock samples were collected in each study area from soil pits, quarries or roadcuts near the plots (maximum distance 15 km). Rock samples from the plots were compared to rock samples from the roadcuts and quarries where possible to ensure that the samples were taken from the same geology.



Each plot was subdivided into four 20 m x 20 m subplots in which soil profiles were sampled in 10 cm increments down to 1 m soil depth and combined to get depth explicit composite samples. We then selected three soil layers for further analyses as they represent distinct sections in a soil profile that differ in C input and biogeochemical soil factors: 0-10 cm (Topsoil, TS), 30-40 cm (Shallow Subsoil, SS) and 60-70 cm (Deep Subsoil, DS). Field moist samples were sieved to 12 mm to get a homogenous substrate still containing the inherent aggregate structure. Samples were then air-dried for 3-5 days. Soil bulk density samples were taken using Kopecky Cylinders. Litter (L) and decomposed organic (O) layers on top of the mineral soil were sampled within a 20 cm x 20 cm square in the center of each subplot and combined to composite samples for L and O layer, respectively.

**2.3 Soil analysis**

A wide range of soil physical and chemical parameters were analyzed in the framework of project TropSOC (Doetterl et al. 2021a;b), from which the following were used in this study as potential covariates for controls on SOC: bulk density, total elements of base cations (Ca, Mg, K, Na), total phosphorus, metal oxides with relevance to C stabilization (Al, Fe, Mn), elements where concentrations relate strongly to weathering (Si, Ti, Zr) and additional soil properties that relate to soil fertility (texture, pH, effective and potential cation exchange capacity, base saturation, bioavailable phosphorus). Generally, each analysis was performed with 20 % of the samples analyzed in triplicates to assess analytical error. Prior to analyses, all samples were oven-dried at 30 °C for 48-72 hours until dry.

*Soil C fractionation and nutrient analysis*

Three soil size fractions representing different stabilization mechanisms associated with varying SOC turnover times were isolated using a microaggregate isolator (Six et al. 2000a; Stewart et al., 2008; Doetterl et al., 2015b). These fractions consisted of: (i) > 250 µm = unprotected C; (ii) 53-250 µm = occluded C in microaggregates; and (iii) < 53 µm = C associated with free silt and clay sized particles (Fig. 2). Briefly, 20 g of 12 mm sieved bulk soil was submerged under water for 24 hours to break up non-water stable aggregates. Next, the slaked soil sample was wet-sieved through a 250 µm sieve using the microaggregate isolator mounted on a sample shaker. The sample was shaken for $20 \pm 11$ min with 50 glass beads to break up any remaining macroaggregates. The remaining material was then wet sieved through a 53 µm sieve by moving the sieve 50 times up and down within two minutes by hand. The isolated soil fractions were then analyzed for carbon and nitrogen (SOC and total nitrogen, TN). To ensure sample homogeneity, the > 250 µm fraction was powdered with a ball mill (Mixer Mill MM 200, Retsch, Germany) prior to C and N analysis. The carbon mass of each soil C fraction ($SOC_{>250µm}$, $SOC_{53-250µm}$ and $SOC_{<53µm}$) was calculated by multiplying the SOC concentration with the corresponding fraction mass. The ratio of the C mass of microaggregate associated C to free silt and clay associated C (m / s+c ratio) was calculated as a proxy to distinguish between soils in which mineral-C protection, which takes place in both fractions, is amplified by the physical protection through aggregation. It is generally interpreted that the higher this ratio, the more C is occluded within stable microaggregates, on top of being stabilized by mineral-organic interactions (and vice versa for low rations). SOC and TN for all samples were analyzed using dry combustion (Vario EL Cube CNS Elementar Analyzer, Germany). The SOC stock of the bulk soil ($SOC_{bulk}$) was calculated by multiplying the SOC concentration with the bulk density and the thickness of the depth increment (10 cm).




*Sequential pedogenic oxide extraction*

To assess the abundance of Al, Fe and Mn bearing phases and their correlation with $SOC_{bulk}$, a three-step sequential extraction of pedogenic oxy-hydroxides (Stucki et al., 1988) was performed on powdered bulk soil (Fig. 2). First, sodium-pyrophosphate at pH 10 was used for extracting organically complexed metals (Bascomb, 1968). Second, ammonium-oxalate-oxalic acid at pH 3 was used for extracting amorphous, short-range order (SRO) secondary oxides and poorly crystalline, alumnosilicates (Dahlgren, 1944). Note that results of pyrophosphate extraction must be interpreted with caution, since Al from Al hydroxide phases and poorly crystalline alumnosilicates can also be partially extracted using this reagent (Schuppli et al., 1983; Kaiser and Zech, 1996). Third, dithionite-citrate-bicarbonate (DCB) at pH 8 was used for extracting crystalline oxy-hydroxides (Mehra and Jackson, 1958). All extracts including the calibration standards were filtered through 41 grade Whatman filters, diluted (1:1000) and then analyzed for elemental concentrations using inductively coupled plasma optical emission spectroscopy (ICP-OES) (5100 ICP-OES Agilent Technologies, USA).

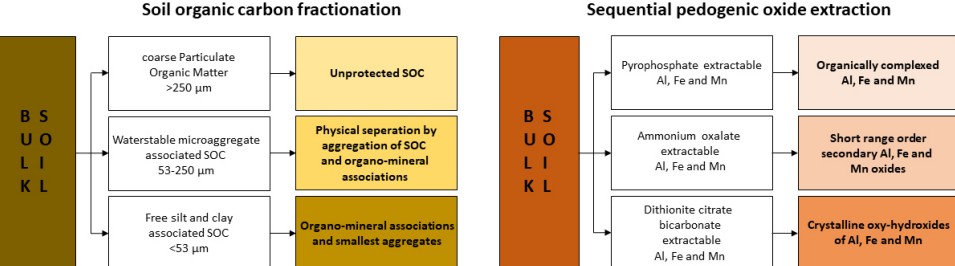

**Figure 2: Left: Applied SOC fractionation scheme after Stewart et al. (2008) and Doetterl et al. (2015b) and its interpretation in terms of functional SOC pools and C stabilization mechanisms. Right: Applied sequential extraction scheme following Stucki et al. (1988) and its interpretation in terms of oxide phases relevant for C stabilization.**

*Calculation of FOC contribution to total $SOC_{bulk}$*

The radioisotopic signature ($\Delta^{14}C$) of bulk soil was assessed using AMS spectrometry at the Max Planck Institute for Biogeochemistry (Jena, Germany) and conventional radiocarbon age following the conventions of Stuiver and Polach (1977). $^{14}C$ radiocarbon dating was used to estimate the relative age differences of C between samples and to estimate the potential contribution of FOC to the total SOC in soils developed from mixed sedimentary rocks. Following Cerri et al. (1985) and Kalks et al. (2020) we assume that the biogenic carbon in the different soil depths of all sites were relatively similar and overall several orders of magnitude younger than the FOC. The baseline values for this assumption are the depth explicit mean values of $\Delta^{14}C$ of the mafic and felsic sites as they are free of FOC. Based on this assumption, the mean depth-specific percent modern carbon (PMC) values for those regions were calculated as follows:

$$f_{bio} = \frac{F}{F_{bio}} * 100 , \qquad (1)$$

Where $F$ is the PMC in a sample, $f_{bio}$ (%) is the proportion of biogenic organic carbon in the total amount of organic carbon, $F_{bio}$ is the fraction PMC averaged from the plateau and slope sites and depths of the mafic and felsic sites.

In a second step, the amount of FOC at the mixed sedimentary site was assessed as follows:



$$f_{FOC} = 100 - f_{bio} , \qquad (2)$$

Where $f_{FOC}$ is the proportion of fossil organic carbon as a fraction of the total amount of soil organic carbon (%).

*Calculation of the chemical index of alteration and elemental differences between parent material and soil*

Based on results of the analyses of total element concentrations (see Doetterl et al., 2021b for details), the chemical index of alteration (CIA %) (Fiantis et al., 2010) was calculated to assess the weathering stage of the soil as follows:

$$CIA = \frac{Al_2O_3}{Al_2O_3 + CaO + Na_2O + K_2O} * 100, \qquad (3)$$

To illustrate gains and losses of nutrients in the soil column compared to the underlying parent material, the relative and absolute difference in element concentration for key elements that enrich or deplete with weathering ($\Delta$Ca, $\Delta$K, $\Delta$Mg, $\Delta$Na, $\Delta$P, $\Delta$Fe, $\Delta$Al, $\Delta$Mn, $\Delta$Si) was assessed using unweathered bedrock samples and soil collected from 30-40 cm depth at the plateau position in each geochemical region. This location and depth  was chosen as it (i) represents the most weathered part of the soil column, (ii) excludes any potential influence by lateral fluxes and (iii) reduced the likelihood of biogenic disturbance through root growth, which concentrates in our sites in organic layers and topsoil (Doetterl et al.,  2021b).

**2.4 Statistical analysis**

The significance level for all statistical analysis was set at $p < 0.05$. Differences with depth, topography and geochemistry of the region for the various SOC fractions and SOC related variables described above were assessed by testing for equality of means using one-way and Welch analysis of variance (ANOVA) (n = 3 for plateaus, n = 6 for slopes and n = 3 for valleys). To avoid type I errors in ANOVA caused by heteroscedasticity (Moder, 2007), we performed Levene's Test (Levene, 1960) for all ANOVAs. Based on the outcome of the Levene's Test, the result of either one-way ANOVA (no heteroscedasticity) or of the Welch ANOVA (heteroscedasticity present) was used. To compare the means of multiple groups, post-hoc pairwise comparison was applied using Bonferroni correction (Day and Quinn, 1989) or Tamhane T2 in the case of unequal variances (Tamhane, 1979).

For dimension reduction of independent potential predictors of SOC, to illustrate the variance of these predictors across the dataset, and to minimize multicollinearity in regression analyses, we performed a varimax-rotated principal component analysis (rPCA) (n = 27) using all non-SOC derived chemical and physical soil variables described above. Only predictor variables with a loading factor of > 0.5 or < -0.5 were interpreted for each rotated component (RC). Because of differences in units and ranges of predictor variables, prior to the rPCA, a Z-score standardization (Lacrose, 2004) was applied as follows:

$$X^* = \frac{X - mean(X)}{SD(X)}, \qquad (4)$$





Where *X\** is the standardized value, *X* is the original value and *SD* is the standard deviation.

Only RCs with an eigenvalue > 1 and explaining > 5 % variances were kept for further statistical analyses. A
mechanistic interpretation of the identified RCs was provided based on the loading of each RC.

The remaining RCs were used as explanatory variables in multiple linear stepwise regressions to the most important predictors explaining differences in SOC variables. We focused our analyses on predicting $SOC_{bulk}$, $\Delta^{14}C$ as well as m / s+c ratios for non-valley positions (n = 27). Valley positions were excluded due to the small
sample size (n = 9). As most of these variables naturally show strong depth trends (Minasny et al., 2016), we added soil depth as an additional explanatory variable in our models to avoid over-interpretation of variables which were cross-correlated to soil depth. After assessing the predictive model strength of our multi RC models on SOC target variables, we assessed the relative importance of explanatory variables using the R-package "Relaimpo" (Grömping, 2006). In a final step, to disentangle the effect of soil depth and RCs to predict our SOC
target variables, partial correlation was used by controlling correlations of RCs to explore whether SOC variables were directly controlled by the rPCs after controlling for soil depth. IBM SPSS Statistics 26 (IBM: SPSS Statistics for Windows, 2019) was used for the ANOVA and partial correlation. The rPCA, regressions and relative importance analysis were realized using R 3.6.1 (R Core Team:, 2020).

**3 Results**

### 3.1 Climate and topography

Note that we have pretested for correlations between SOC stocks, mean annual temperature (MAT) and mean annual precipitation (MAP) across our study sites. No significant correlations were found with the included climatic variables (data not shown) indicating no significant effect of climatic variation between sites on SOC
dynamics. Hence, we focused our further analyses on the impact of local geochemistry and topography on SOC stocks and stabilization.

For all tested SOC variables, significant differences in the means of different topographic positions within each geochemical region were found between valley and non-valley positions (plateaus and slopes) with higher SOC stocks in valley positions compared to non valley positions. No significant differences were found between plateau
and slope positions (Table A1). Even though valley positions are of the same geochemistry as the non-valley positions, geochemical soil properties in valleys were significantly different than at non valley positions, as fluvial activity and sedimentation unrelated to hillslope processes were dominant (See Supplement 1_additional short results and discussion for valley positions). Consequently, for all follow-up analyses on differences with soil depth and geochemistry, the dataset was split into valley positions versus non-valley positions. Due to the limited sample
size for the valley positions (n = 1-3), no further statistical analysis was applied. However, all valley data is available in a short supplementary result and discussion section attached to this manuscript (Supplement 1).



### 3.2 Soil chemical weathering stage and pedogenic oxides

*Parent material geochemistry and weathering stage*

Parent material, from which soils in the three geochemical regions have developed, showed distinct differences in elemental composition (Doetterl et al., 2021a; b), with generally low concentrations of Ca, Mg and Na base cations (0.01 - 0.58 mass%). Al and Fe concentration were significantly higher in the mafic (Al: $6.27 \pm 2.84$; Fe: $8.98 \pm 1.84$ mass%) than sedimentary rocks region (Al: $0.62 \pm 0.41$; Fe: $2.32 \pm 1.73$ mass%) and the felsic region (Al: $0.52 \pm 1.21$; Fe: $1.09 \pm 1.58$ mass%). Similarly, total P was highest in the mafic region (P: $0.37 \pm 0.14$ mass%)

compared to the mixed sedimentary (P: $0.02 \pm 0.02$ mass%) and the felsic region (P: $0.01 \pm 0.1$ mass%). In contrast, Si content was lowest in the mafic region (Si: $14.22 \pm 2.01$ mass%) compared to the mixed sedimentary rocks region (Si: $36.11 \pm 7.01$ mass %) and the felsic region (Si: $37.29 \pm 5.92$ mass%).

Across geochemical regions, soils were highly weathered as indicated by high CIA values of 78 - 99 % at all three soil depths (data not shown). Soils developed from mafic parent material were depleted in Ca, Mg and Na base

cations compared to parent material (Ca: -38 to -100 %; Mg: -72 to -87 %; Na: -90 to 375 %). Soils developed in the mixed sedimentary region were depleted in Ca ($-100\pm172$ %) compared to parent material but not for Mg or Na. Soils developed from felsic parent material showed a substantial increase in all base elements (Ca: up to 1240 %; Mg: up to 1015 %; Na: up to 677 %). All soils were enriched in Al compared to parent material (mafic: up to 307 %; felsic: up to 6859 %; mixed sedimentary: up to 1514 %). Similarly, all soils were enriched in Fe compared

to parent material (mafic: up to 85 %; felsic: up to 1482 %; mixed sedimentary: up to 3486 %). Soils developed from mafic parent material showed depletion in P (-72 to 14 %) compared to the parent material. In contrast, we observed an enrichment in P for soils developed from mixed sedimentary (up to +2583 %) and felsic parent material (up to 6671 %) compared to parent material. Note that the extraordinarily high differences in P between soil and parent material in the latter regions are mainly related to the fact that P concentration in parent material

of the felsic and sediment region were extremely small to begin with, but did accumulate in soil through fixation in the the biosphere and and plant uptake (Wilcke et al., 2002; Wang et. al., 2010). This interpretation is supported by the observations that P mass in all three geochemical regions for the investigated soil layers used for comparison to parent material converge (mafic: $0.15 \pm 0.002$ mass%; felsic: $0.05 \pm$ N/A mass%; mixed sedimentary: $0.08 \pm 0.01$ mass%) while other less critical elements for biological processes leached during soil

development. For example, all soils were depleted in Si compared to the parent material (mafic: -27 to 8 %; felsic: -62 to -26 %; mixed sedimentary: -71 to -58 %), which is indicative of long-term weathering.

*Pedogenic oxides*

For non-valley positions, pyrophosphate extractable oxides (0.02 to 1.93 mass%) and oxalate extractable oxides

(0.32 to 2.33 mass%) were low compared to DCB extractable oxides (2.74 to 13.63 mass%) (Fig. 3). Pyrophosphate extractable oxides showed no significant differences across geochemical regions. Oxalate extractable oxides were significantly higher across all soil depths in mafic (1.68 to 2.33 mass%) and mixed sedimentary (0.32 to 2.28 mass%) soils compared to soils developed from felsic parent material (0.35 to 0.91 mass%). DCB extractable oxides across all soil depths were significantly higher in mafic soils (9.54 to 13.63

mass%) compared to soils developed from felsic (2.74 to 7.03 mass%) and mixed sedimentary parent material (3.55 to 8.44 mass%).





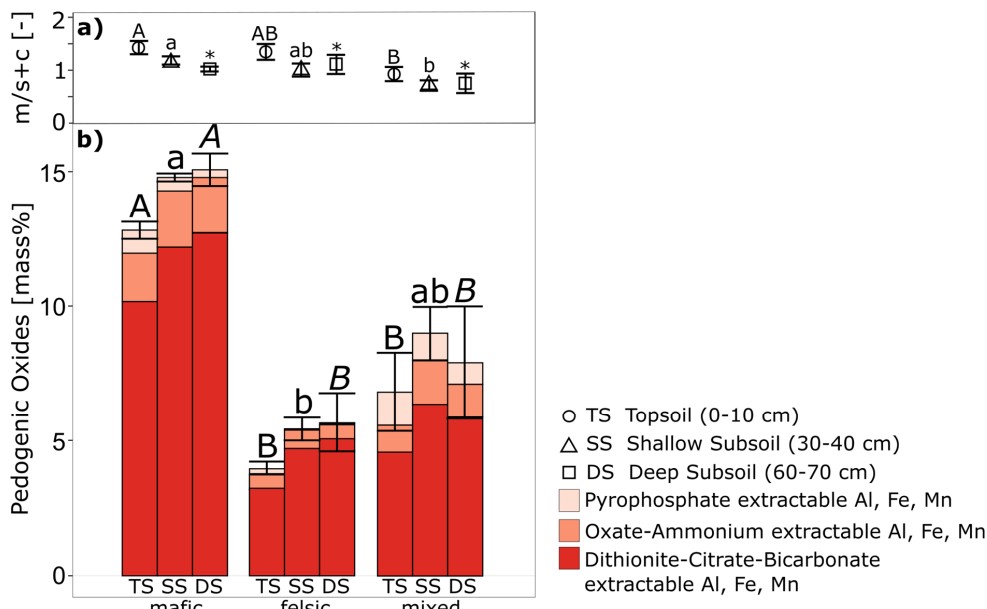

**Figure 3: (a) m / s+c ratio (n = 9 per bar) and (b) pedogenic oxide fractions (n = 3 per bar) of the sequential extraction across geochemical regions in non-valley positions. Letters indicate significant differences between geochemical regions per soil depth for m / s+c ratio (a) and total pedogenic oxide mass (b). Asterisks indicate no significant differences in means (p > 0.05). Error bar represents standard error.**

### 3.3 Variation in SOC properties with geochemistry

*SOC$_{bulk}$*

SOC$_{bulk}$ in topsoil did not differ across geochemical regions (mafic: 50.6 ± 13.9 t C ha$^{-1}$; felsic: 45.3 ± 3.9 t C ha$^{-1}$; mixed sedimentary: 44.0 ± 3.9 t C ha$^{-1}$) whereas subsoil SOC$_{bulk}$ was significantly smaller in the felsic region (shallow subsoil: 16.3 ± 2.5 t C ha$^{-1}$; deep subsoil: 10.1 ± 1.9 t C ha$^{-1}$) compared to the mafic (shallow subsoil: 28.3 ± 3.3 t C ha$^{-1}$; deep subsoil: 22.9 ± 7.2 t C ha$^{-1}$) and mixed sedimentary region (shallow subsoil: 30.4 ± 2.8 t C ha$^{-1}$; deep subsoil: 23.9 ± 4.7 t C ha$^{-1}$). Note that while SOC$_{bulk}$ decreased strongly with depth in the mafic and felsic region, only a weak decrease of SOC$_{bulk}$ with depth was observed in the mixed sedimentary region (Fig. 4).



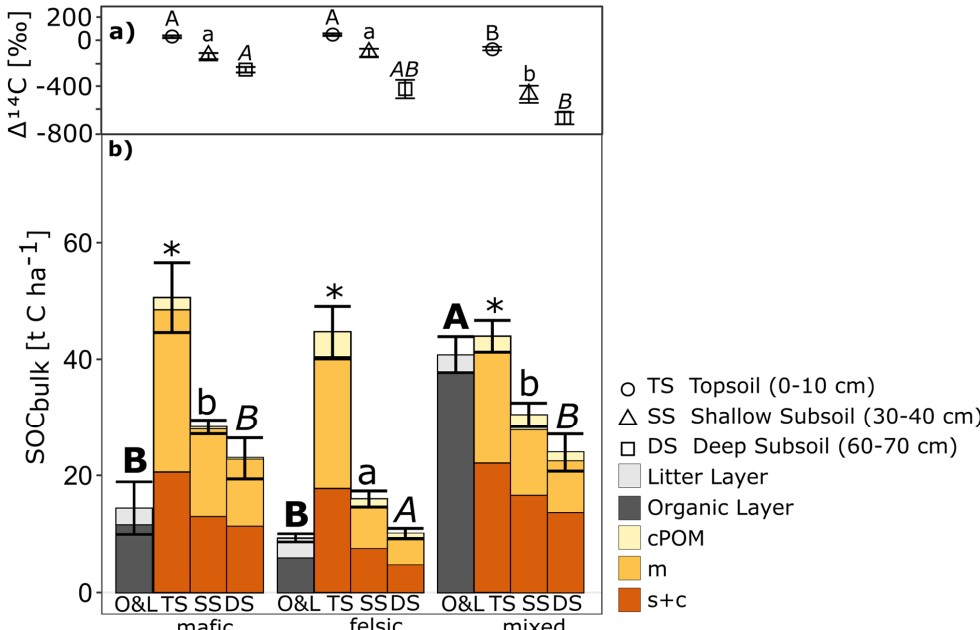

**Figure 4: (a) $\Delta^{14}$C across geochemical regions in non-valley positions (n = 3 per bar). (b) SOC$_{bulk}$ and fractions across geochemical regions in non-valley positions (n = 9 per bar). Letters indicate significant differences between geochemical regions per soil layer for $\Delta^{14}$C (a) and SOC$_{bulk}$ (b). Asterisks indicate no significant differences in means (p > 0.05). For $\Delta^{14}$C values, error bars are smaller than symbols. cPOM – coarse particulate organic matter, m – waterstable microaggregates, s+c – free silt and clay fraction.**

*Abundance of C fractions*

At all sites and soil depths, fractions were dominated by microaggregate associated C (SOC$_{53-250\mu m}$), contributing 32.6 ± 9.9 % to 55.2 ± 1.4 % of SOC$_{bulk}$, and free silt and clay associated C (SOC$_{<53\ \mu m}$) contributing 40.3 ± 4.8 % to 54.5 ± 2.4 % of SOC$_{bulk}$. Coarse particulate organic carbon (SOC$_{>250\ \mu m}$) contributed to 1.4 ± 0.2 % to 11.1 ± 1.5 % of SOC$_{bulk}$. Microaggregate associated C and m / s+c ratios were generally higher in topsoils (SOC$_{53-250\mu m}$: 43.19 ± 6.77 % to 55.23 ± 1.43 %; m / s+c: 0.93 ± 0.17 to 1.42 ± 0.04) compared to subsoils (SOC$_{53-250\mu m}$: 32.64 ± 9.88 % to 49.65 ± 1.84 %; m / s+c: 0.75 ± 0.2 to 1.07 ± 0.35) (Figure 3a). For topsoil and shallow subsoil, m / s+c ratio was significantly higher in the mafic region than in the sediment region, meaning more microaggregate associated C in mafic than in mixed sediments compared to free s+c associated C. The felsic region ranges between the other two geochemical regions in this regard and was not significantly different to either one. Deep subsoil m / s+c indicated the same trends but was not statistically different across regions (Fig. 3a). Note that the relative abundance of microaggregate associated and free silt and clay associated with C was not significantly different with topographic positions (Table A1).

*Changes in $^{14}$C signature*

Soils from all geochemical regions at non-valley positions were significantly more depleted in $\Delta^{14}$C with increasing soil depth (Fig. 4a, 4b). The $\Delta^{14}$C ranged from 31.2 ± 11.5 ‰ (mafic) to -78.7 ± 26.6 ‰ (mixed sedimentary) in the topsoil, and -257.8 ± 32.8 ‰ (mafic) to -675.2 ± 89.6 ‰ (mixed sedimentary) in subsoil. While there were no significant differences between $\Delta^{14}$C of comparable samples of the felsic and mafic region,



their counterparts from the mixed sedimentary region were significantly more depleted in $\Delta^{14}C$. The contribution
of FOC to soil C in the mixed sedimentary region increased significantly with soil depth for non-valley positions,
ranging from $11.3 \pm 2.6$ % FOC in topsoils to $52.0 \pm 13.2$ % in subsoils (Table 1).

**Table 1: Proportion of biogenic vs. fossil derived organic carbon (OC) in soils developed from mixed
sedimentary rocks in non-valley positions (n = 3). FOC values in felsic and mafic soils = 0 % (data not
shown).**

| depth increment | amount of biogenic OC (%) | amount of fossil OC (%) |
|---|---|---|
| 0 - 10 cm | $88.7 \pm 2.6$ | $11.3 \pm 2.6$ |
| 30 - 40 cm | $60.7 \pm 14.5$ | $39.3 \pm 14.5$ |
| 60 - 70 cm | $48.0 \pm 13.2$ | $52.0 \pm 13.2$ |

**3.4 Rotated principal component explained variance and loadings**

Four rPCs were determined ($rPC_{nv}$) explaining 78.1 % of the variance in the non-valley position subset (Fig. 5;
Table A3). $rPC1_{nv}$ (Eigenvalue 9.34, explaining 33.4 % of the variance) represents solid phase mineralogy, in
which total metal oxide concentration ($\sum$ Fe, Al, Mn), DCB extractable oxide concentration and the Al / Si ratio
had strong positive loadings (variable loading > 0.9), while Si, pH / clay ratio and sand content had strong negative
loadings (< - 0.7). $rPC2_{nv}$ (Eigenvalue 9.15, explaining 32.7 % of the variance) represents the chemistry of the
soil solution where exchangeable bases, CEC base saturation / clay ratio and Ca / Ti ratio showed strong positive
loadings ($\geq 0.87$) and exchangeable acidity, CIA and $SOC_{organic}$ showed strong negative loadings (< - 0.5). $rPC3_{nv}$
(Eigenvalue 1.82, explaining 6.5 % of the variance) represents silt content and the C stock of organic layers
($SOC_{organic}$), both having a strong negative loading (< -0.5). $rPC4_{nv}$ (Eigenvalue 1.57, explaining 5.6 % of the
variance) represents organo-mineral complexes with pyrophosphate extractable oxides concentration showing a
strong positive loading (< 0.8).





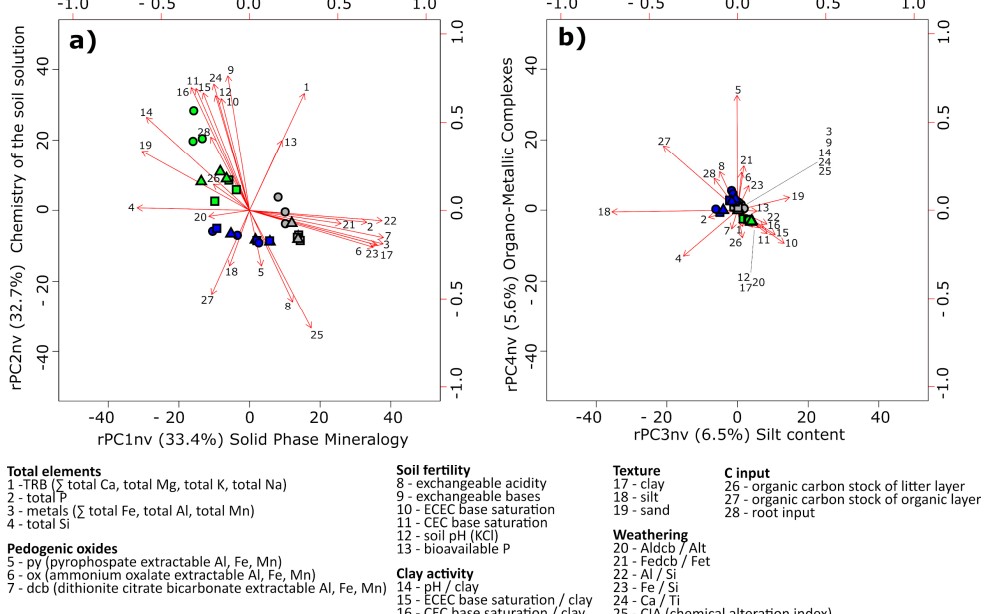

**Figure 5:** Biplots of the varimax rotated principal component analysis. (a) rPC1$_{nv}$ and rPC2$_{nv}$ and (b) rPC3$_{nv}$ and rPC4$_{nv}$ of non-valley positions (n = 27). Observations cluster together based on similarities within geochemical regions and their distinction to other geochemical regions. Vector length indicates how strongly variables influence a specific rPC. The angles between vectors display the degree of auto-correlation between variables. Small angles represent positive correlations and high degree of autocorrelation, diverging angles represent negative correlations and high degree of autocorrelation, high angles indicate no correlations between variables and/or rPCs.

### 3.5 Explained variability and relative importance of predictors (non-valley soils)

Soil depth and rPC4$_{nv}$ explained 73 % of variability ($R^2$) in SOC$_{bulk}$ (p < 0.01). Soil depth contributed 82 % to the explanatory power of the model (Table 2). The second most important explanatory variable in our model was rPC4$_{nv}$, which represented organo-mineral complexes (p < 0.01) and contributed 18 % to the explanatory power of the model. Soil depth and rPC1$_{nv}$, rPC3$_{nv}$, and rPC4$_{nv}$, could explain 75 % of variability ($R^2$) in $\Delta^{14}$C data (p < 0.01). Soil depth contributed 75 % to the explanatory power of the model, followed by rPC3$_{nv}$ (silt content, 16 % explanatory power), rPC4$_{nv}$ (organo-mineral complexes, 5 % explanatory power) and rPC1$_{nv}$ (solid phase mineralogy, 4 % explanatory power). Soil depth, rPC1$_{nv}$, and rPC2$_{nv}$ could explain 44 % of variability ($R^2$) in m / s+c ratio (p < 0.01). rPC2$_{nv}$ contributed 46 % to the explanatory power of the model, followed by rPC1$_{nv}$ (solid phase mineralogy, 31 % explanatory power), and soil depth (23 % explanatory power), making m / s+c ratios the only SOC target variable not highly correlated to soil depth.



**Table 2: Multiple linear stepwise regression analysis (beta coefficients) and relative importance analysis in brackets for non-valley soils. $SOC_{bulk}$, $\Delta^{14}C$ and m / s+c ratio are explained by soil depth and the extracted rPCs. \*p <0.1; \*\*p <0.05; \*\*\*p <0.01. Adjusted $R^2$ displays the goodness of fit and the root mean square error (RMSE) assesses the model quality.**

| response | soil depth | $rPC1_{nv}$ - solid phase mineralogy | $rPC2_{nv}$ - chemistry of the soil solution | $rPC3_{nv}$ - silt content | $rPC4_{nv}$ - organo-mineral complexes | adjusted $R^2$ | RMSE |
|---|---|---|---|---|---|---|---|
| $SOC_{bulk}$ | -0.77\*\*\* (82 %) | | | | 0.31\*\*\* (18 %) | 0.73 | 0.46 |
| $\Delta^{14}C$ | -0.87\*\*\* (75 %) | 0.39\*\* (4 %) | | 0.24\* (16 %) | -0.32\* (5 %) | 0.75 | 0.47 |
| m / s+c | -0.29\* (23 %) | 0.74\*\*\* (31 %) | 0.8\*\*\* (46 %) | | | 0.44 | 0.39 |

### 3.6 Partial correlations controlled for soil depth

When controlling for soil depth, correlations between $SOC_{bulk}$ and the identified rPCs became significant with changes in correlation coefficients ranging from 0.01 to 0.44 (Table 3). Solid phase mineralogy explained 16.8 %, chemistry of the soil solution 15.2 %, silt content 16.8 % and organo-mineral complexes 28.1 % of the variability ($R^2$) in $SOC_{bulk}$. Correlation between $\Delta^{14}C$ and the chemistry of the soil solution became insignificant and declined by 0.13, whereas the correlation of silt content and organo-mineral complexes improved by 0.16 and 0.22, respectively. Silt content explained 32.5 % and organo-mineral complexes explained 12.3 % of the variability ($R^2$) in $\Delta^{14}C$. Correlation between the m / s+c ratio and chemistry of the soil solution decreased by 0.09 with only silt content left with a significant correlation. Silt content explained 22.1 % of the variability ($R^2$) in m / s+c ratio.

**Table 3: Partial correlation analysis between SOC variables ($SOC_{bulk}$, $\Delta^{14}C$ and m / s+c ratio) and extracted rPCs controlling for soil depth. Zero-order correlation displays the Pearson r when including no control variable. The controlled correlation shows the Pearson r when controlling for soil depth. \*p <0.1; \*\*p <0.05; \*\*\*p <0.01. Pearson r and $R^2$ derive from simple linear regression between the individual rPCA and the respective SOC variable. Dark red: r > 0.4, light red: r > 0.3, light blue: r < -0.3, dark blue: r < -0.4.**

| response | control | $rPC1_{nv}$ - solid phase mineralogy Pearson r | $R^2$ | $rPC2_{nv}$ - chemistry of the soil solution Pearson r | $R^2$ | $rPC3_{nv}$ - silt content Pearson r | $R^2$ | $rPC4_{nv}$ - organo-mineral complexes Pearson r | $R^2$ |
|---|---|---|---|---|---|---|---|---|---|
| $SOC_{bulk}$ | zero-order | 0.06 | 0.00 | 0.05 | 0.00 | -0.19 | 0.04 | 0.41\*\* | 0.17 |
| | soil depth | 0.41\*\* | 0.17 | -0.39\*\* | 0.15 | -0.41\*\* | 0.17 | 0.53\*\*\* | 0.28 |
| $\Delta^{14}C$ | zero-order | -0.05 | 0.00 | 0.44\*\* | 0.19 | 0.41\*\* | 0.17 | -0.13 | 0.02 |
| | soil depth | 0.19 | 0.04 | 0.31 | 0.10 | 0.57\*\*\* | 0.32 | -0.35\* | 0.12 |
| m / s+c | zero-order | 0.13 | 0.02 | 0.38\* | 0.14 | 0.46\*\* | 0.21 | -0.22 | 0.05 |
| | soil depth | 0.23 | 0.05 | 0.29 | 0.08 | 0.47\*\* | 0.22 | -0.30 | 0.09 |



## 4 Discussion

### 4.1 Soil C stabilization driven by soil chemistry and parent material

In contrast to our initial hypothesis that topography affects C stabilization in tropical forest soils through lateral material movements, we found no indication of this in our analysis (Supplementary results and short discussion therein). Despite prolonged chemical weathering, parent material leaves an identifiable, long-lasting footprint in the chemical properties of tropical forest soils (Fig. 5). Overall, the differences in elemental composition between parent materials in each geochemical region, together with enrichment and depletion processes of elements during weathering, has resulted in soils with specific properties and prerequisites for SOC stabilization (Table A2). Especially stabilization mechanisms related to pedogenic oxides (Fig. 3) and the formation of organo-mineral associations are relevant for SOC stabilization at our sites, as illustrated by the strong correlation of variables representing organo-mineral complexes to $SOC_{Bulk}$ and $\Delta^{14}C$ (Table 3). The influence of parent material geochemistry and weathering on the pattern of physically separated soil C fractions, in particular on stabilizing C in microaggregates, was smaller across soil geochemical regions than across soil depths. (Figs. 3, 4 and Table 3). The most important variables for explaining m / s+c ratios were found to be soil depth, solid phase mineralogy and the chemistry of the soil solution which could in total explain 44 % of m / s+c variance (Table 2). We interpret the high m / s+c ratios in topsoils as indicative for the formation of stable microaggregates promoted by the higher abundance of C, which functions as a binding agent (Denef and Six, 2005) and the generally more fertile conditions in tropical topsoil compared to subsoil favouring microbial activity (Kidinda et al., 2021). The abundance of pedogenic oxides further promotes aggregation by providing reactive mineral surfaces (Oades, 1988). In this study, pedogenic oxides are determined by geochemistry with higher contents in mafic compared to felsic and mixed sedimentary soils (Fig. 3b). Therefore, mafic soils also stabilize more C in microaggregates. It is likely that the high amount of pedogenic oxides usually measured in microaggregates (Doetterl et al., 2015a) and the low amount of POM measured in our study overall suggests that predominantly mineral-complexed SOC accumulated within the isolated aggregates. This is supported by studies showing that microaggregate sized particles in deeply weathered tropical soils are rich in Fe and Al concretions (Cooper et al., 2005; Zotarelli et al., 2005; Denef et al. 2007; Martinez and Souza, 2019). In the light of this finding, aggregation is an important means to promote the complexation of C with minerals (Martinez and Souza, 2019), but also the result of the tendency of pedogenic oxides to form stable aggregates (Doetterl et al., 2015a), which lends additional protection of soil C. When controlling for soil depth, our geochemical predictors gained or retained similar prediction power (Table 3), showing the importance for predicting SOC target variables at all soil depths.

### 4.2 Fossil organic carbon contributions to $SOC_{bulk}$ and driving $\Delta^{14}C$

While depth trends in $\Delta^{14}C$ were similar across geochemical regions (Fig. 4a), soils developed on mixed sedimentary rocks were significantly depleted in $\Delta^{14}C$ for all topographic positions. A significant amount between $11.3 \pm 2.6$ to $52.0 \pm 13.2$ % FOC was found to contribute to $SOC_{bulk}$ in soils developed on mixed sedimentary rocks (Table 1), thus supporting our initial hypothesis (iii) that FOC bearing parent material will strongly impact $SOC_{bulk}$. Despite contributions of FOC of up to 52 %, $SOC_{bulk}$ did not differ to the same extent between the geochemical regions (Fig. 4b). Two potential explanations could be found for this observation. First, fertility conditions and stabilization mechanisms in soils developed from mixed sedimentary parent material reduces the



amount of SOC with modern biogenic origin drastically due to slower C cycling (Trumbore, 2009). However,
neither $\Delta^{14}$C nor the analyzed distribution of soil C fractions support this explanation. Biologically active topsoil
$\Delta^{14}$C in the mixed sediment region was less depleted than the subsoil counterparts when being compared across
regions (Figure 4).  Additionally, auxiliary data on the overall net primary productivity of the investigated systems
(Doetterl and Fiener, 2021b) point towards relatively comparable C inputs across the three regions at least in
topsoil. Subsoils of the sediment region, however, may receive significantly less C input, which is the subject of
future investigations. A second explanation could be the decomposition of FOC once it enters more biologically
active zones (i.e. topsoils), where climatic and edaphic conditions are more suitable for microbial decomposer
communities. Findings on $\Delta^{14}$C signatures of respired C indicate the presence of FOC contributing to $CO_2$ release
(Bukombe et al., 2021). Here, on average only $6.7 \pm 2.5$ % of the respired $CO_2$ showed a fossil origin in the non-
valley positions. Hence, the fact that FOC content increases with depth (Table 1) but is nearly depleted in topsoils
indicates that these sources of fossil C, even though a poor source of nutrients and energy for microorganisms
(Hemmingway et al., 2018), can become available to microbial decomposition under more suitable conditions.
Statistically, differences in $\Delta^{14}$C were best explained by soil depth (Table 2), but between regions by the presence
of FOC in soils developed from mixed sedimentary parent material, and when controlling for soil depth with
geochemical variables (Table 3). Further, our model identified silt content as a strong predictor for $\Delta^{14}$C. It is
possible that tropical soils form very stable silt-sized microaggregates (Six et al., 2000b), in which FOC is
potentially stabilized. However, the low pedogenic oxide content in the mixed sedimentary region, important for
microaggregate formation (Zotarelli et al., 2005; Denef et al., 2007; Doetterl et al., 2015a; Martinez and Souza,
2020), is not entirely supportive of this interpretation. There is also no statistically significant relationship between
fine soil texture classes and the C associated with microaggregates or the free silt and clay fraction in our
investigated sites in the sediment region (data not shown). We argue, therefore, that no mechanistic relationship
exists between $\Delta^{14}$C and silt content, and the observed relationship is to be interpreted as an autocorrelation
between the high amount of $^{14}$C depleted FOC in the mixed sedimentary region and the fact that these sediments
have a higher silt content than their felsic and mafic counterparts.

**4.3 Interpreting soil controls for predicting SOC dynamics**

Our regression analyses revealed that a wide variety of soil variables contributes to predicting SOC and its
turnover in a quantitative and qualitative way (Table 2 & A3). An exact mechanistic interpretation is difficult due
to the relatively small number of observations compared to potential predictor variables. However, in general a
set of variables related to soil fertility, and the chemistry of the solid phase and soil solution contributed to
predicting our three target variables: 1) $SOC_{bulk}$, 2) $\Delta^{14}$C, and 3) m / s+c (Table 2 & A3). Furthermore, some
interpretation of the included rotated components is possible because their respective loading is clearly distinct
from each other (Table A3, Figure 5). Notably, the prediction power of our models was dominated by soil depth
(Table 2) for all three target variables. However, partial correlation revealed that soil depth covered relationships
between the target variables and our predictors (Table 3), indicating that soil depth is autocorrelated to variability
in soil mineralogy and soil fertility. For example, solid phase mineralogy and organo-mineral complexes reflect
the amount of total elements in the soil and their transformation into secondary minerals and thus the amount of
reactive mineral surfaces, which are highly relevant in the sorptive protection of SOC (Oades, 1984; Evanko and
Dzombak, 1998; Kleber et al., 2015). Similarly, the chemistry of the soil solution represents properties that are





relevant for C stabilization. Here, low pH levels can mobilize $Al^{3+}$, which eventually sorbs onto reactive mineral

surfaces preventing C stabilization (Smith, 1999). In our mafic soils, sorptive C stabilization created by pedogenic oxides leads to high $SOC_{bulk}$ (Fig. 3, 4 and Table 2) and supports the formation of aggregates. Conversely, felsic soils are low in pedogenic oxides and thus have low sorption potential and consequently also the lowest $SOC_{bulk}$. Clay content, identified as a major factor for stabilizing SOC in temperate soils (Angst et al., 2018) and also in geomorphologically active tropical soil systems (Quesada et al., 2020) was not identified as a major control for

our soils. This illustrates the importance of understanding soil geochemical preconditions when identifying controls of C dynamics and that findings are not necessarily transferable, even between comparable soil types and climates. Overall, we found that chemical stabilization of SOC, especially by organo-mineral complexation, contributed the most to explaining differences in $SOC_{bulk}$ in the analyzed soils while aggregation, profiting from the abundance of pedogenic oxides and stable Fe & Al structures in soils added additional C stabilization potential.

Hence, under similar climatic conditions and similar C input through vegetation (Doetterl et al., 2021b), our data indicates that C stabilization mechanisms in soil control SOC stocks and turnover in deeply tropical weathered soils more so than soil fertility conditions and drive patterns of SOC stocks across geochemical regions (Figures 3, 4). Our results are comparable to other studies investigating the impact of reactive mineral surfaces on $SOC_{bulk}$ in tropical kaolinitic soils (Bruun et al., 2010) and the importance of sorptive processes to C on mineral surfaces

(Jagadamma et al., 2014). While these processes have been identified to be important for soil C stabilization in a variety of ecosystems and climate zones (Murphy et al., 1992; Evanko and Dzombak, 1998; Kögel-Knabner et al., 2018; Fang et al., 2019), they stand out in tropical soils due to the high amounts of Fe and Al oxides in the system overall and the advanced stage of weathering of soils that led to the formation of low activity clays and other end member minerals with low potential for C sorption (Ito and Wagai, 2017).


## 5 Conclusions and outlook

Overall, we found a minimal impact of topography on SOC variables as a function of soil fluxes along slopes, but observed higher SOC stocks in valleys compared to non-valley positions due to varying hydrological conditions and alluvial processes. Instead, chemical soil properties, derived from parent material geochemistry, were

identified as major explanatory factors. We argue that the strong role of geochemical variables in explaining SOC is a function of reactive mineral surfaces dependent on the composition of the parent material and its weathering status. More available reactive surfaces will favour sorptive C stabilization and the formation of stable aggregates, thus leading to higher $SOC_{bulk}$. In the investigated deeply weathered tropical soil systems, the formation of organo-mineral complexes of Al, Fe and Mn were most important for explaining $SOC_{bulk}$ across geochemical regions and

the impact of clay content was minimal. Differences in the relative abundance of C associated with microaggregates and with free silt and clay fractions differed significantly between geochemical regions and soil depth, indicating that despite long lasting weathering, mafic soils can protect C better than their felsic und mixed sediment counterparts. Aggregate formation in tropical soils seems to profit from the abundance of pedogenic oxides in soils, linking two of the most important mineral related C stabilization mechanisms in soil with

geochemical variability retained from parent material. Differences in $\Delta^{14}C$ were best explained with soil depth and the presence of FOC, which appears to be decomposable by microbial communities under more fertile, topsoil conditions. It is recommended to repeat our analyses in other tropical soil and land use systems, where external drivers such as soil redistribution, weathering stages or stabilization mechanisms might differ. This way, a more





spatially explicit picture of C stabilization mechanisms will contribute to understanding future tropical SOC

dynamics in light of ongoing climatic and land use changes, as well as the representation of SOC stabilization and

destabilization in land surface models.

## 6 Appendices

**Table A1: SOC variables grouped by soil depth, topographic position and geochemical regions (Section 1: Mafic; Section 2: Felsic; Section 3: Mixed); Different capital letters indicating significant differences in means (p < 0.05). Asterisks indicate no significant differences in means (p > 0.05). Means are compared across topographical positions for each depth increment in each geochemical region (plateau n = 3, slopes n = 6, valley n = 3).**

### Section 1: mafic magmatic

| parameter | unit | 0 - 10 cm | | | 30 - 40 cm | | | 60 - 70 cm | | |
|---|---|---|---|---|---|---|---|---|---|---|
| | | plateau | sloping | valley | plateau | sloping | valley | plateau | sloping | valley |
| C | mass% | 8.54 ± 3.37 * | 5.4 ± 0.59 * | 8.62 ± 1.11 * | 2.11 ± 0.02 * | 2.55 ± 0.32 * | 3.8 ± 0.66 * | 1.42 ± 0.27 * | 2.02 ± 0.92 * | 2.62 ± 0.72 * |
| N | mass% | 0.7 ± 0.19 * | 0.55 ± 0.06 * | 0.87 ± 0.12 * | 0.21 ± 0.01 A | 0.25 ± 0.02 A | 0.38 ± 0.05 B | 0.15 ± 0.01 * | 0.2 ± 0.1 * | 0.25 ± 0.04 * |
| C / N | - | 11.78 ± 1.89 * | 9.75 ± 0.68 * | 9.92 ± 0.78 * | 10.17 ± 0.23 * | 10.26 ± 0.76 * | 10.14 ± 1.24 * | 9.61 ± 0.99 * | 10.18 ± 0.92 * | 10.52 ± 2.32 * |
| $SOC_{bulk}$ | tC / ha | 65.63 ± 25.9 * | 43.07 ± 6.6 * | 62.24 ± 11.41 * | 25.02 ± 0.24 A | 29.98 ± 3.22 A | 41.22 ± 4.6 B | 17.2 ± 3.2 * | 25.78 ± 12.06 * | 32.19 ± 9.15 * |
| m / s+c | - | 1.43 ± 0.3 * | 1.41 ± 0.45 * | 1.72 ± 0.83 * | 1.3 ± 0.08 * | 1.13 ± 0.28 * | 1.66 ± 0.5 * | 1.11 ± 0.11 * | 0.98 ± 0.15 * | 1.66 ± 1.06 * |
| $SOC_{>250\ \mu m}$ | % | 5.09 ± 2.37 * | 3.62 ± 1.14 * | 4.12 ± 3.38 * | 1.48 ± 0.33 * | 1.36 ± 0.42 * | 2.18 ± 1.57 * | 1.32 ± 0.03 * | 1.31 ± 1.01 * | 2.12 ± 1.04 * |
| $SOC_{53-250\ \mu m}$ | % | 55.3 ± 3.38 * | 55.2 ± 9.49 * | 57.9 ± 12.83 * | 55.63 ± 1.59 * | 51.52 ± 7.26 * | 60.28 ± 7.44 * | 51.73 ± 2.54 * | 48.61 ± 3.83 * | 56.98 ± 16.71 * |
| $SOC_{<53\ \mu m}$ | % | 39.61 ± 5.75 * | 41.18 ± 8.52 * | 37.98 ± 13.96 * | 42.89 ± 1.47 * | 47.12 ± 6.93 * | 37.54 ± 6.03 * | 46.95 ± 2.51 * | 50.08 ± 3.76 * | 40.91 ± 15.67 * |

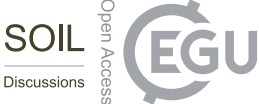

**Section 2: felsic magmatic**

| parameter | unit | 0 - 10 cm | | | 30 - 40 cm | | | 60 - 70 cm | | |
|---|---|---|---|---|---|---|---|---|---|---|
| | | plateau | sloping | valley | plateau | sloping | valley | plateau | sloping | valley |
| C | mass% | 4.58 ± 1.14 * | 3.39 ± 0.79 * | 2.99 ± 0.39 * | 0.95 ± 0.07 * | 0.87 ± 0.21 * | 0.55 ± 0.13 * | 0.55 ± 0.08 * | 0.51 ± 0.12 * | 0.28 ± 0.11* |
| N | mass% | 0.45 ± 0.08 * | 0.35 ± 0.07 * | 0.29 ± 0.03 * | 0.1 ± 0.01 * | 0.09 ± 0.02 * | 0.06 ± 0.01 * | 0.06 ± 0.01 * | 0.05 ± 0.01 * | 0.03 ± 0.02* |
| C / N | - | 10.18 ± 0.65 * | 9.77 ± 0.62 * | 10.31 ± 1.98 * | 10.29 ± 0.04 * | 10.31 ± 1.81 * | 10.37 ± 0.52 * | 9.57 ± 0.04 * | 10.6 ± 2 * | 9.89 ± 1.55 * |
| $SOC_{bulk}$ | tC / ha | 46.75 ± 28.49 * | 44.08 ± 9.72 * | 40.42 ± 8.03 * | 16.41 ± 2.87 * | 15.81 ± 4.61 * | 9.88 ± 2.25 * | 9.03 ± 1.05 * | 10.32 ± 2.96 * | 5.95 ± 3.26 * |
| m/ s +c | - | 1.23 ± 0.54 AB | 1.37 ± 0.47 B | 2.45 ± 0.39 A | 0.68 ± 0.01 * | 1.11 ± 0.37 * | 0.72 ± 0.07 * | 1.06 ± 0.28 * | 1.13 ± 0.62 * | 0.53 ± 0.13 * |
| $SOC_{>250 \mu m}$ | % | 12.77 ± 6.31 * | 10.15 ± 3.84 * | 12.57 ± 7.94 * | 5.62 ± 2.66 * | 9.43 ± 9.97 * | 3.71 ± 1.6 * | 3.41 ± 0.47 * | 8.26 ± 7.71 * | 4.88 ± 2.37 * |
| $SOC_{53-250 \mu m}$ | % | 46.52 ± 6.42 * | 50.49 ± 7.8 * | 61.96 ± 7.91 * | 38.14 ± 1.41 * | 46.34 ± 9.4 * | 40.16 ± 2.9 * | 49.31 ± 6.29 * | 45.34 ± 15.37 * | 32.94 ± 5.73* |
| $SOC_{<53 \mu m}$ | % | 40.71 ± 12.73 AB | 39.36 ± 8.45 A | 25.47 ± 2.23 B | 56.24 ± 1.25 * | 44.22 ± 9.51 * | 56.13 ± 1.47 * | 47.28 ± 6.75 * | 46.4 ± 14.02 * | 62.18 ± 3.38 * |





**Section 3: mixed sedimentary rocks**

| parameter | unit | 0 - 10 cm | | | 30 - 40 cm | | | 60 - 70 cm | | |
|---|---|---|---|---|---|---|---|---|---|---|
| | | plateau | sloping | valley | plateau | sloping | valley | plateau | sloping | valley |
| C | mass% | 6.95 ± 0.46 * | 4.83 ± 1.91 * | 6.3 ± 3.05 * | 2.42 ± 0.15 * | 2.41 ± 1.06 * | 1.63 ± 0.78 * | 1.36 ± 0.26 * | 1.78 ± 0.86 * | 2.28 ± 2.14 * |
| N | mass% | 0.47 ± 0.06 * | 0.31 ± 0.13 * | 0.37 ± 0.18 * | 0.15 ± 0.01 * | 0.14 ± 0.07 * | 0.06 ± 0.03 * | 0.09 ± 0.01 * | 0.09 ± 0.07 * | 0.06 ± 0.05 * |
| C / N | - | 14.84 ± 0.98 * | 15.85 ± 1.86 * | 17.04 ± 0.73 * | 15.63 ± 1.46 A | 18.63 ± 4.71 A | 27.15 ± 2.98 B | 15.55 ± 0.99 * | 33.41 ± 37.42 * | 30.29 ± 17.28 * |
| SOC$_{bulk}$ | tC / ha | 48.41 ± 6.59 * | 41.71 ± 8.47 * | 52.75 ± 30.72 * | 31.08 ± 3.33 * | 30.06 ± 7.42 * | 26.88 ± 15.82 * | 20.2 ± 4.26 * | 25.74 ± 11.29 * | 40.86 ± 38.35 * |
| m / s+c | - | 0.9 ± 0.66 * | 0.94 ± 0.31 * | 2.97 ± 2.43 * | 0.53 ± 0.08 * | 0.8 ± 0.34 * | 1.12 ± 0.59 * | 0.85 ± 0.2 * | 0.7 ± 0.69 * | 1.15 ± 0.59 * |
| SOC$_{>250\,\mu m}$ | % | 6.74 ± 3.98 * | 6.43 ± 7.14 * | 22.19 ± 12.29 * | 13.37 ± 19.44 * | 6.26 ± 1.72 * | 15.09 ± 9.18 * | 4.41 ± 3.8 * | 6.37 ± 4.6 * | 10.42 ± 11.4 * |
| SOC$_{53-250\,\mu m}$ | % | 40.18 ± 15.39 * | 44.7 ± 9.46 * | 52.71 ± 6.82 * | 30.22 ± 8.01 * | 40.16 ± 9.54 * | 42.34 ± 6.19 * | 43.28 ± 3.52 * | 27.31 ± 20.91 * | 44.56 ± 9.71 * |
| SOC$_{<53\,\mu m}$ | % | 53.07 ± 17.58 * | 48.87 ± 6.1 * | 25.11 ± 14.87 * | 56.4 ± 12.28 * | 53.59 ± 8.99 * | 42.57 ± 13.79 * | 52.3 ± 7.28 * | 49.65 ± 30.66 * | 45.02 ± 18.14 * |






**Table A2: Absolute and relative differences of mineral soil at plateau positions (30 - 40 cm soil depth) compared to parent material. Maximum and minimum value refers to the maximum and minimum depletion respectively. If sample size was n = 1, the corresponding standard deviation was not applicable (N/A).**

| | | mafic | | felsic | | mixed | |
|---|---|---|---|---|---|---|---|
| | | soil absolute difference [ppm] | relative difference Δ [%] | soil absolute difference [ppm] | relative difference Δ [%] | soil absolute difference [ppm] | relative difference Δ [%] |
| Ca | MEAN ± SD | -5765 ± 5814 | -99 ± 101 | 1443 ± N/A | 1240 ± N/A | -55 ± 95 | -100 ± 172 |
| | MAX | -12937 | -100 | 1028 | 194 | -164 | -100 |
| | MIN | -75 | -38 | 1559 | N/A | 0 | N/A |
| K | MEAN ± SD | 161 ± 970 | 20 ± 603 | 1148 ± N/A | 618 ± N/A | -248 ± 566 | -32 ± 228 |
| | MAX | -1621 | -65 | 743 | 126 | -774 | -62 |
| | MIN | 984 | 2687 | 1318 | 8790 | 499 | 306 |
| Mg | MEAN ± SD | -10414 ± 3417 | -83 ± 33 | 1160 ± N/A | 1015 ± N/A | 587 ± 211 | 527 ± 36 |
| | MAX | -14333 | -87 | 718 | 129 | 341 | 179 |
| | MIN | -5488 | -72 | 1274 | N/A | 876 | 2235 |
| Na | MEAN ± SD | -519 ± 850 | -63 ± 164 | 148 ± N/A | 677 ± N/A | 43 ± 59 | 92 ± 138 |
| | MAX | -1954 | -90 | 102 | 149 | -34 | -53 |
| | MIN | 313 | 375 | 163 | 2414 | 117 | 834 |
| P | MEAN ± SD | -2133 ± 1361 | -58 ± 64 | 403 ± N/A | 746 ± N/A | 585 ± 201 | 289 ± 34 |
| | MAX | -3774 | -72 | 223 | 96 | 333 | 89 |
| | MIN | 186 | 14 | 450 | 6671 | 874 | 2583 |
| Fe | MEAN ± SD | 28849 ± 25496 | 32 ± 88 | 30867 ± N/A | 284 ± N/A | 87917 ± 24429 | 379 ± 28 |
| | MAX | -6967 | -6 | -12878 | -24 | 56114 | 157 |
| | MIN | 60099 | 85 | 39115 | 1482 | 121456 | 3486 |
| Al | MEAN ± SD | 24730 ± 37287 | 39 ± 151 | 25226 ± N/A | 487 ± N/A | 49252 ± 5537 | 799 ± 11 |
| | MAX | -33553 | -32 | -9013 | -23 | 41728 | 383 |
| | MIN | 78824 | 307 | 29966 | 6859 | 55961 | 1514 |
| Mn | MEAN ± SD | 2040± 1461 | 160 ± 72 | 1251 ± N/A | 3126 ± N/A | 168 ± 48 | 776 ± 29 |
| | MAX | 584 | 34 | 1125 | 675 | 91 | 168 |
| | MIN | 3273 | 308 | 1291 | N/A | 213 | 17591 |
| Si | MEAN ± SD | -16700 ± N/A | -12 ± N/A | -216070 ± N/A | -58 ± N/A | -241033 ± N/A | -67 ± N/A |
| | MAX | -46000 | -27 | -256000 | -62 | -298700 | -71 |
| | MIN | 9500 | 8 | -55900 | -26 | -163000 | -58 |




**Table A3: Rotated principal components and variable loadings of varimax rotated principal component analysis for non-valley positions (rPC) (n = 27). Variable loadings of pearson r > 0.5 and < - 0.5 are highlighted. The following variables contributed to the components: To describe the total element content in the soils (associated with $rPC1_{nv}$ and $rPC4_{nv}$) the total reserve in base cations (TRB; sum of total Mg, Ca, K and Na), total P, metals (sum of total Fe, Al and Mn) and total Si were used. Pyrophosphate, oxalate and DCB extractable phases of Al, Fe and Mn characterised the pedogenic oxides. Exchangeable acidity and bases, ECEC basse saturation, CEC base saturation, $pH_{KCL}$ and bio available P (bray-P) were reflecting soil fertility (associated with $rPC2_{nv}$). The ratio of pH / clay, ECEC base saturation / clay and CEC base saturation / clay were proxies for clay activity (associated with $rPC2_{nv}$). Clay, silt and sand described the texture (associated with $rPC1_{nv}$ and $rPC3_{nv}$) and several weathering indices ($Al_{dcb}$ / $Al_t$ ratio, $Fe_{dcb}$ / $Fe_t$ ratio, Al / Si ratio, Fe / Si ratio, Ca / Ti ratio and the chemical index of alteration CIA) characterised the soil weathering stage (predominantly associated with $rPC1_{nv}$). $SOC_{litter}$, $SOC_{organic}$ and root input were describing the C input (associated with $rPC2_{nv}$ and $rPC3_{nv}$).**

|  |  |  | $rPC1_{nv}$ | $rPC2_{nv}$ | $rPC3_{nv}$ | $rPC4_{nv}$ |
|---|---|---|---|---|---|---|
|  | Eigenvalue |  | 9.34 | 9.15 | 1.82 | 1.57 |
|  | Proportion Var (%) |  | 0.33 | 0.33 | 0.07 | 0.06 |
|  | Cumulative Var (%) |  | 0.33 | 0.66 | 0.73 | 0.78 |
|  | **Mechanistic interpretation** |  | **solid phase mineralogy** | **chemistry of the soil solution** | **silt content** | **organo-mineral complexes** |
|  | **Total elements** |  |  |  |  |  |
| 1 | TRB | mass% | 0.39 | **0.83** | ~0 | -0.12 |
| 2 | total P | mass% | **0.83** | ~0 | -0.21 | ~0 |
| 3 | metals ($\sum$ total Fe, total Al, total Mn) | mass% | **0.95** | -0.24 | ~0 | ~0 |
| 4 | total Si | mass% | **-0.80** | ~0 | -0.38 | -0.32 |
|  | **Pedogenic oxides** |  |  |  |  |  |
| 5 | pyro. extr. Fe, Al, Mn | mass% | ~0 | -0.39 | ~0 | **0.82** |
| 6 | oxalate extr. Fe, Al, Mn | mass% | **0.88** | -0.26 | ~0 | 0.31 |
| 7 | dcb extr. Fe, Al, Mn | mass% | **0.95** | -0.19 | ~0 | -0.13 |
|  | **Soil fertility** |  |  |  |  |  |
| 8 | exchangeable acidity | me / 100g | 0.31 | **-0.65** | -0.13 | 0.27 |
| 9 | exchangeable bases | me / 100g | -0.15 | **0.96** | ~0 | ~0 |
| 10 | ECEC base saturation | % | -0.20 | **0.79** | 0.33 | -0.23 |
| 11 | CEC base saturation | % | -0.38 | **0.87** | 0.21 | -0.17 |
| 12 | soil pH (KCl) | - | -0.24 | **0.82** | 0.21 | ~0 |
| 13 | bio available P (bray) | mg kg$^{-1}$ | 0.23 | **0.50** | 0.14 | ~0 |
|  | **Clay activity** |  |  |  |  |  |
| 14 | pH / clay | - | **-0.73** | **0.66** | ~0 | ~0 |
| 15 | ECEC base saturation / clay | - | -0.33 | **0.83** | 0.27 | -0.17 |
| 16 | CEC base saturation / clay | - | -0.41 | **0.87** | 0.15 | -0.12 |



**Continuation of Table A3**

|  |  |  |  |  |  |  |
|---|---|---|---|---|---|---|
|  | **Texture** |  |  |  |  |  |
| 17 | clay | % | **0.89** | -0.24 | ~0 | ~0 |
| 18 | silt | % | -0.14 | -0.39 | **-0.89** | ~0 |
| 19 | sand | % | **-0.76** | 0.42 | 0.37 | ~0 |
|  | **Weathering** |  |  |  |  |  |
| 20 | $Al_{dcb} / Al_t$ | - | -0.29 | ~0 | 0.11 | -0.12 |
| 21 | $Fe_{dcb} / Fe_t$ | - | **0.65** | ~0 | ~0 | 0.27 |
| 22 | Al / Si | - | **0.94** | ~0 | 0.19 | -0.12 |
| 23 | Fe / Si | - | **0.91** | -0.25 | ~0 | 0.17 |
| 24 | Ca / Ti | - | -0.25 | **0.90** | ~0 | ~0 |
| 25 | CIA | % | 0.44 | **-0.84** | ~0 | ~0 |
|  | **C input** |  |  |  |  |  |
| 26 | $SOC_{litter}$ | t C ha$^{-1}$ | -0.26 | 0.19 | ~0 | -0.20 |
| 27 | $SOC_{organic}$ | t C ha$^{-1}$ | -0.27 | **-0.59** | **-0.53** | 0.46 |
| 28 | root input | kg m$^{-3}$ | -0.28 | **0.52** | -0.17 | 0.23 |









**7 Data availability**

All data used in this study will be published in an open access project-specific database with a separate DOI (Doetterl et al., 2021b). The specific data of this publication is available upon request from the corresponding author.

**8 Sample availability**

All soil samples are logged and barcoded at the Department of Environmental Science at ETH Zurich, Switzerland.

**9 Team list**

See acknowledgments and author list.

**10 Author contributions**

SD and PF designed the research. MR conducted the sampling campaign and collected the data. All authors analyzed and interpreted the data. All authors contributed to the writing of the paper.

**11 Competing interests**

SD is a liaison editor of the special issue *Tropical biogeochemistry of soils in the Congo Basin and the African Great Lakes region* and JS is an executive editor and PF is a topical editor of the SOIL journal. However, none of them were involved in the review process of this manuscript. All other authors declare that they have no conflict of interest.

**12 Special issue statement**

This article is part of the Special Issue: *Tropical biogeochemistry of soils in the Congo Basin and the African Great Lakes region*.

**13 Acknowledgements**

This work is part of the DFG funded Emmy Noether Junior Research Group "Tropical soil organic carbon dynamics along erosional disturbance gradients in relation to soil geochemistry and land use" (TROPSOC; project number 387472333). The authors like to thank following collaborators of this project: International Institute of Tropical Agriculture (IITA), Max Planck Institute for Biogeochemistry, Institute of Soil Science and Site Ecology at Technical University Dresden, Sustainable Agroecosystems Group and the Soil Resources Group both located at ETH Zurich and the Faculty of Agriculture at the Catholic University of Bukavu. The authors like to thank the whole TROPSOC team especially the student assistants for their important work in the laboratory and all field work helpers making the sampling campaign possible.



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
