# Peer review of "The role of geochemistry in organic carbon stabilization against microbial decomposition in tropical rainforest soils"

_SOIL, 2020_

## Referee Comment (RC3)

Dear editor and authors,

I really appreciate the opportunity to read and comment on the manuscript Soil-2020-92 entitled "The role of geochemistry in organic carbon stabilization in tropical rainforest soils". Overall, the manuscript is very interesting and covers a topic of global interest, and as such, it would fulfill the requirements for publication in your Journal. I would recommend accepting, although a few minor issues should be dealt with first. I acknowledge the fact that other reviewers participated in the discussion and gave detailed suggestions to improve the original version of the manuscript. For the version of the text I have read, there are no major issues that would require extensive reworking. So, only minor issues will be pointed out in the following text and I hope my comments can still be useful.

**Abstract**

Lines 22-23. I could not find strong evidence to support the claim that "fluvial dynamics and changed hydrological conditions had a secondary control on SOC dynamics in valley positions, leading to higher SOC stocks there than at the non-valley positions". This should be better explained. How can the reader agree that "fluvial dynamics and changed hydrological conditions" can be inferred from the results reported and help to explain higher SOC stocks in valleys than in non-valley positions?

Lines 23-24. I believe the term "Fossil organic carbon" could be more precisely described and referred to as "geogenic organic carbon" in the whole manuscript. In fact, geogenic organic carbon was used by the authors themselves elsewhere in their text (e.g., page 3, line 116).

**Introduction, Section 1.2. Environmental and geochemical controls on SOC dynamics in tropical forests**

Lines 69-70. To what extent geogenic carbon (FOC) is preserved owing to its inherent chemical properties (e.g., recalcitrance) or the specific conditions (e.g., burial of organic carbon mixed with mineral particles) under sedimentary environments?

Line 86. Please provide a reference in which the authors have reported pedogenic oxides contents above 50% in the clay fraction of tropical soils. I agree that some tropical soils can exhibit more than 50% of pedogenic oxides, but such soils are not the norm as implied in the text. Please check.

**Introduction, Section 1.4 Study aims**

Line 140. "composition" or "decomposition"?

**Hypotheses**

All hypotheses proposed should be rephrased and put into simpler functional relationships e.g., $y = f(x)$. This would reduce verbosity and give the reader a glimpse on how each hypothesis would be effectively tested. I believe the hypotheses can be better used to guide the reader through the Discussion section as well.

Hypothesis (i): what parameters would be used/measured to determine the control of topography on lateral fluxes of water and mineral mass? This is not clear to me.

Hypothesis (ii): I understood the context, but verbosity can be reduced.

Hypothesis (iii): I found the third hypothesis particularly confusing as it includes a reference to "priming effects", which were not measured in this study.
Example to rephrase the third hypothesis:
iii) Geogenic soil carbon stocks vary more consistently as a function of soil depth than landscape position or soil parent material.

**Results, Section 3.1 Climate and topography**
I believe the supplement 1 could benefit the reader if kept in the main text, all results reported therein are very nice. Besides, as far as I understood hypothesis (i), the observation of higher soil C stocks in valleys than in non-valley positions is important for this research.
Lines 350-353. The inference that "Even though valley positions are of the same geochemistry as the non-valley positions, geochemical soil properties in valleys were significantly different than at non valley positions, as fluvial activity and sedimentation unrelated to hillslope processes were dominant", seems quite speculative to explain higher C stocks in valleys relative to non-valley positions. In my opinion, a predominant effect of "fluvial and sedimentation" rather than "hillslope processes" would make sense only if the geochemistry in the valleys were significantly different from that observed in non-valley positions.

**Results, Section 3.3 Variation in SOC properties with geochemistry**
Lines 410-411. In the sentence "Note that while SOCbulk decreased strongly with depth in the mafic and felsic region, only a weak decrease of SOCbulk with depth was observed in the mixed sedimentary region (Fig. 4)", can we infer that SOC buildup followed the accumulation of sediments over time to a greater extent than C inputs from the local vegetation? How does the $^{14}$C depth-trend compare to that observed in valley positions as shown in Fig. S2?

**Results, Section 3.5 Explained variability and relative importance of predictors (non-valley soils)**
Lines 471-472. Based on the observation that "Soil depth and rPC4$_{nv}$ explained 73 % of variability ($R^2$) in SOCbulk (p < 0.01). Soil depth contributed 82 % to the explanatory power of the model", how (in)sensitive tropical C pools may be to changes in climate or land use?

**Results, Section 3.6 Partial correlations controlled for soil depth**
It looks quite amazing that when the effect of depth is controlled, the explanatory power of the other variables included in the model does not increase substantially (except silt content). How does this trend compare to temperate ecosystems? What can be inferred about the relationship between pedogenesis and soil C accumulation in the tropics? What is the mineralogy of the silt fraction? Given the data shown in Table 3 and Figure 5, such information is very important for this research and would facilitate the discussion (lines 570-578).

**Discussion, Section 4.1 Soil C stabilization driven by soil chemistry and parent material**
To what extent the inference that "In contrast to our initial hypothesis that topography affects C stabilization in tropical forest soils through lateral material movements, we found no indication of this in our analysis (Supplementary results and short discussion therein)" can be reconciled

with the observation of higher SOC stocks in valley positions, despite exhibiting similar geochemistry to non-valley positions?

**Conclusions and outlook**

Lines 410-412. "Differences in $\Delta^{14}$C were best explained with soil depth and the presence of FOC, which appears to be decomposable by microbial communities under more fertile, topsoil conditions." There is an apparent redundancy here since $\Delta^{14}$C would co-variate with FOC and factors limiting microbial respiration at depth should be more important than soil fertility.

---

## Author Comment (AC1)

**Point-by-point response to community comment by Dr. Dan Wan**

Dear Dr. Dan Wan,

First of all, thank you very much for taking the time to read and comment on our manuscript. Please see the following point-by-point response to your important comments. Reviewer original comments are highlighted in grey.

We hope you find our response and changes to the manuscript satisfying and we are looking forward to hearing your opinion on our revised manuscript.

Yours sincerely,

The authors

*COMMUNITY COMMENT 1: "1) this is a research article, so the subtitle of the introduction is unnecessary, the authors should rearrange this section"*

**Our response:** Thanks for the comment. According to the Journal's guidelines, we decided to use subtitles in the introduction to provide better guidance to the reader, since it helps to assimilate the key points of the study framework much faster in our opinion. We also wanted to be consistent with the subtitle structure throughout the manuscript. Nevertheless, if the editors are sharing the same opinion that removing the subtitles helps to streamline the introduction, we will happily do so.

*COMMUNITY COMMENT 2: "2) line 260: the concentration of OM-complexed metal usually very low, diluted this solution by 1000 folds, the concentration of Fe, Al and Mn may below the detection limit of ICP-OES."*

**Our response:** Thanks for pointing this out. The concentration of pedogenic oxides (Al, Fe, Mn), especially of DCB-extractable oxides, was high in the majority of our samples. We were expecting such high concentrations of pedogenic oxides, since we sampled Ferralsols and Nitisols. We used 5 calibration points using a multielement standard solution. The concentration of each calibration point was at least two-times above the detection limit of the ICP-OES. To cover the concentration range of all our analyzed samples with the above mentioned calibration points, the dilution ratio of 1:1000 worked best. We also used internal standards (i.a. of other Ferralsols) to check the accuracy of the extraction, dilution and measuring procedure. The average error/deviation of 3.8±12.03 rel. % was small, therefore giving us confidence in the accuracy of the ICP-OES data.

*COMMUNITY COMMENT 3: "3) the SOC stabilization mechanism was pH-dependent (Rasmussen et al., 2018 Biogeochemistry 137, 297– 306; Wan et al., 2019 Eur. J. Soil Sci. 70, 1153–1163; Wan et al., 2021 Sci. Total Environ. 10.1016/j.scitotenv.2021.145037), therefore, the soil pH should be added to the results and discussed."*

**Our response:** Thank you very much for pointing out this very interesting literature. Indeed, our data shows a significant relationship between pyro and oxalate extr. oxides (py+ox) with soil pH. Comparable to the study of Rasmussen et al. (2018), we observed higher oxide contents with lower soil pH (Figure R1). However, soil pH was not a significant predictor for py+ox. Instead, the effect of the geochemical region on py+ox was highly significant (Table R1). We conclude that the amount of highly relevant C stabilization partners represented by py+ox is dependent on the geochemical region. Hence, we focus on the pedogenic oxides fractions in the result section. Furthermore, we accounted for the effect of soil pH on soil properties and SOC stocks by including it in the rotated principal component analysis. For

more detailed information about other geochemical soil properties (i.a. soil pH), we kindly refer to the project-specific database publication (Doetterl et al., 2021a; 2021b).

[Figure]

**Figure R1: pyro. and oxalate extr. oxides (Al, Fe, Mn) in relation to soil pH (KCl). Including non-valley positions (n = 27).**

**Table R1: Multiple linear regression explaining content of pyro. and oxalate extr. oxides (Al, Fe, Mn) with soil pH (KCl) and geochemical region (dummy coded). Including non-valley positions (n = 27).**

|  | Unstandardized Coefficients (B) | Std. Error | Standardized Coefficients (beta) | t | Sig. |
|---|---|---|---|---|---|
| (Constant) | 1.751 | 1.005 |  | 1.743 | 0.095 |
| soil pH (KCl) | 0.222 | 0.274 | 0.167 | 0.809 | 0.427 |
| geochemistry = felsic | -2.12 | 0.48 | -0.95 | -4.418 | 0 |
| geochemistry = mixed | -0.223 | 0.335 | -0.1 | -0.665 | 0.513 |

response: py+ox
reference: geochemistry = mafic
ANOVA: p < 0.001
Adj. R²: 0.61

*Used literature:*

Doetterl, S., Asifiwe, R.K., Baert, G., Bamba, F., Bauters, M., Bukombe, B., Cadisch, G, Cizungu, L., Cooper, M., Hoyt, A., Kabaske, C., Kalbitz, K., Kidinda, K.L., Maier, A., Mainka, M., Mayrock, J., Muhindo, D., Mujinya, B., Mukotanyi, S.M., Nabahungu, L., Reichenbach, M., Rewald, B., Six, J., Stegmann, A., Summerauer, L., Unseld, R., van Oost, K., Verheyen, K., Vogel, C., Wilken, F., Fiener, P. Organic matter cycling along geochemical, geomorphic and disturbance gradients in forests and cropland of the African Tropics - Project TropSOC DATABASE_v1.0, Earth System Science Data DISCUSSIONS (pre-print), https://doi.org/10.5194/essd-2021-73, 2021a.

Doetterl, S.; Bukombe, B.; Cooper, M.; Kidinda, L.; Muhindo, D.; Reichenbach, M.; Stegmann, A.; Summerauer, L.; 36 Wilken, F.; Fiener, P. TropSOC Database. Version 1.0. GFZ Data Services. https://doi.org/10.5880/fidgeo.2021.009, 37, 2021b.

Rasmussen, C., Heckman, K., Wieder, W. R., Keiluweit, M., Lawrence, C. R., Berhe, A. A., Blankinship, J. C., Crow, S. E., Druhan, J. L., Pries, C. E. H., Marin-Spiotta, E., Plante, A. F., Schädel, C., Schimel, J. P., Sierra, C. A., Thompson, A., and Wagai, R.: Beyond clay: towards an improved set of variables for predicting soil organic matter content, Biogeochemistry, 137, 297-306, https://doi.org/10.1007/s10533-018-0424-3, 2018.

*COMMUNITY COMMENT 4: "4) the meaning of letters (e.g. a, b, A, B) in figures 3 and 4 was not clear, I can not understand these figures easily. Besides, the color scheme of all figures was not appropriate, the authors should select the same color system or use grayscales (white, black, and gray) or patterns for bars. Table 3 has the same problem."*

**Our response:** Thanks for the comment. The ANOVA compare means across geochemical regions for each soil layer. Each soil layer has its own letter font to guide the reader which bars are compared with each other. We pointed this out in more detail in the Figure caption (Figure 4 example 1). The authors think that the color scheme of Figure 3 and 4 helps to quickly differentiate between soil C fractions and pedogenic oxide fractions without having to read the full figure caption. Using only grayscales will make it difficult to differentiate between the different soil layers and fractions (Figure 4 example 1). Another option would be to use the colorblind safe color scheme from colorbrewer2.org (Figure 4 example 2) or using the recommendations of Fortuna et al. (2013) (Figure 4 example 3). If the editors share the same opinion with Dr. Dan Wan, we will revise the color scheme of the figures and also that of table 3.

[Figure]

**Figure 4 example 1: (a) m / s+c ratio (n = 9 per bar) and (b) pedogenic oxide fractions (n = 3 per bar) of the sequential extraction across geochemical regions in non-valley positions. Letters sharing the same text font indicate significant differences between geochemical regions per soil depth for m / s+c ratio (a) and total pedogenic oxide mass (b). Asterisks indicate no significant differences in means (p > 0.05). Error bar represents standard error.**

[Figure]

**Figure 4 example 2**

[Figure]

**Figure 4 example 3**

*Used Literature*

Fortuna, S. L. J., Kulkarni, C., Stone, M., and Heer, J.: Selecting semantically-resonant colors for data visualization, EuroVis, 32, 401-410, 10.1111/cgf.12127, 2013.

We hope we have addressed all concerns and look forward to hearing from you.

 Best regards,

The authors

---

## Author Comment (AC2)

**Point-by-point response to anonymous Referee#1 comments**

Dear Referee#1,

We would like to thank you for your time and thorough evaluation of our manuscript "*The role of geochemistry in organic carbon stabilization in tropical rainforest soils*", (https://doi.org/10.5194/soil-2020-92). We are very pleased that you positively assessed our work and recognized its relevance. Your comments helped us to significantly improve our manuscript and we want to sincerely thank you for the constructive and valuable insights.

We have addressed all comments and suggestions to the best of our ability. Please find below a point-by-point response to all the concerns raised and how we addressed them. Reviewer original comments are highlighted in grey. New text to be added or modified in the manuscript has quotation marks and is blue-colored in the response.

We hope you find our response and changes to the manuscript satisfying and we are looking forward to hearing from you.

Yours sincerely,

The authors

*REVIEWER#1 COMMENT 1: "The authors use the term "organic carbon stabilization" in the title and in many other parts of the manuscript. It could be stabilization against decomposition, temperature, erosion, dispersion, oxidation, or all these together. However, the authors do not provide a context in which the word "stability" is used."*

**Our response:** We agree with the reviewer that the term "organic carbon stabilization" needs more context. In our manuscript, we use this term as "carbon stabilization against microbial decomposition" Our study and the project it is implemented in aims to understand soil microbial activity, C cycling and C stabilization against decomposition in tropical soils and their feedback between geology, geomorphology and pedogenesis (Doetterl et al. 2021a; 2021b). In this context, we focus on C stabilization against microbial decomposition in the manuscript. We have accordingly changed this term throughout the manuscript where appropriate and also in the title:

"The role of geochemistry in organic carbon stabilization against microbial decomposition in tropical rainforest soils."

*Used literature:*

Doetterl, S., Asifiwe, R.K., Baert, G., Bamba, F., Bauters, M., Bukombe, B., Cadisch, G, Cizungu, L., Cooper, M., Hoyt, A., Kabaske, C., Kalbitz, K., Kidinda, K.L., Maier, A., Mainka, M., Mayrock, J., Muhindo, D., Mujinya, B., Mukotanyi, S.M., Nabahungu, L., Reichenbach, M., Rewald, B., Six, J., Stegmann, A., Summerauer, L., Unseld, R., van Oost, K., Verheyen, K., Vogel, C., Wilken, F., Fiener, P. Organic matter cycling along geochemical, geomorphic and disturbance gradients in forests and cropland of the African Tropics - Project TropSOC DATABASE_v1.0, Earth System Science Data DISCUSSIONS (pre-print), https://doi.org/10.5194/essd-2021-73, 2021a.

Doetterl, S.; Bukombe, B.; Cooper, M.; Kidinda, L.; Muhindo, D.; Reichenbach, M.; Stegmann, A.; Summerauer, L.; 36 Wilken, F.; Fiener, P. TropSOC Database. Version 1.0. GFZ Data Services. https://doi.org/10.5880/fidgeo.2021.009, 37, 2021b.

*REVIEWER#1 COMMENT 2: "In the introduction (section 1.2 Environmental and geochemical controls on SOC dynamics in tropical forests), you never talk about very important environmental*

*controls such as temperature, moisture, pH, redox potential, and oxygen diffusion. You should explore the optimum conditions (e.g., temperature and moisture) for enzymatic activity in the tropics, which ultimately will determine organic matter decomposition rates"*

**Our response:** The authors agree that it is necessary to highlight the importance of environmental conditions on enzymatic activity in the tropics, since it is an important component in C cycling. We therefore changed the paragraph accordingly:

"Climatic factors such as temperature and precipitation are also strong drivers of soil environmental conditions, which can greatly influence soil microbial activity and hence C mineralization and turnover (Davidson and Janssens, 2006; Zhang et al., 2011; Feng et al., 2017). For example, decomposition rates increase in general with temperature, but soil microbial communities adapted to high temperatures are less sensitive to warming (Blagodatskaya et al., 2016). Microorganisms in humid tropical soils with varying soil moisture content are exposed to fluctuating redox potentials but remain active (DeAngelis et al., 2010). C-depleted tropical subsoils contain small but metabolically active microbial communities contributing to C cycling (Kidinda et al., 2020; Stone et al., 2014). Low soil pH in combination with high clay content dominated by pedogenic oxides show a high potential to stabilize enzymes on mineral surfaces which will affect microbial C acquisition (Dove et al., 2020; Allison and Vitousek, 2005; Liu et al., 2020). These climate-driven factors also influence SOC dynamics but more indirectly through the interaction with soil geochemical factors (Doetterl et al., 2015). "

*Used Literature:*

Allison, S.D., Vitousek, P. M.: Responses of extracellular enzymes to simple and complex nutrient inputs, Soil Biol. Biochem., 37, 937–944, https://doi.org/10.1016/j.soilbio.2004.09.014, 2005.

Blagodatskaya, E., Blagodatsky, S., Khomyakov, N., Myachina, O., and Kuzyakov, Y.: Temperature sensitivity and enzymatic mechanisms of soil organic matter decomposition along an altitudinal gradient on Mount Kilimanjaro, Scientific Reports, 6, 10.1038/srep22240, 1-11, 2016.

Davidson, E. A., and Janssen, I. A.: Temperature sensitivity of soil carbon decomposition and feedbacks to Climate Change, Nature, 440, 165-173, https://doi.org/10.1038/nature04514, 2016.

DeAngelis, K. M., Silver, W. L., Thompson, A. W., and Firestone, M. K.: Microbial communities acclimate to recurring changes in soil redox potential status, Environ. Microbio., 12, 3137-3149, doi:10.1111/j.1462-2920.2010.02286.x, 2010.

Doetterl, S., Stevens, A., Six, J., Merckx, R., van Oost, K., Pinto, M. C., Casanova-Katny, A., Muñoz, C., Boudin, M., Venegas, E. Z., and Boeckx, P.: Soil carbon storage controlled by interactions between geochemistry and climate, Nat. Geosci., 8, 780–783, https://doi.org/10.1038/ngeo2516, 2015.

Dove, N. C., Arogyaswamy, K., Billings, S. A. Botthoff, J. K. Carey, C. J. Caitlin, C., DeForest, J. L., Dawson, F., Fierer, N., Gallery, R. E., Kaye, J. P., Lohse, K. A., Maltz, M. R., Mayorga, E., Pett-Ridge, J., Yang, W. H., Hart, S. C., Aronson, E. L.: Continental-scale patterns of extracellular enzyme activity in the subsoil: an overlooked reservoir of microbial activity, Environ. Res. Lett., 15, 1040a1, https://doi.org/10.1088/1748-9326/abb0b3, 2020.

Feng, W., Liang, J., Hale, L. E., Jung, C. G., Chen, J., Zhou, J., Xu, M., Yuan, M., Wu, L., Bracho, R., Pegoraro, E., Schuur, E. A. G., and Luo, Y.: Enhanced decomposition of stable soil organic carbon and microbial catabolic potentials by long-term field warming, Global Change Biol., 23, 4765–4776, https://doi.org/10.1111/gcb.13755, 2017.

Kidinda, L. K., Olagoke, F. K., Vogel, C., Kalbitz, K., and Doetterl, S.: Patterns of microbial processes shaped by parent material and soil depth in tropical rainforest soils, SOIL DISCUSSIONS, in review, https://doi.org/10.5194/soil-2020-80, 2020.

Liu, J., Chen, J., Chen, G., Guo, J., Li, Y.: Enzyme stoichiometry indicates the variation of microbial nutrient requirements at different soil depths in subtropical forests, PLoS ONE 15, e0220599, https://doi.org/10.1371/journal.pone.0220599, 2020.

Stone, M. M., DeForest, J. L., and Plante, A. F.: Changes in extracellular enzyme activity and microbial community structure with soil depth at the Luquillo Critical Zone Observatory, Soil Biology & Biochemistry, 75, 237-247, http://dx.doi.org/10.1016/j.soilbio.2014.04.017, 2014.

Zhang, L., Zeng, G., and Tong, C.: A review on the effects of biogenic elements and biological factors on wetland soil carbon mineralization, Acta Ecol. Sin., 31, 5387-5395, 2011.

REVIEWER#1 COMMENT 3: *"It is not clear in your hypothesis ii, the proxy for the amount of "stabilized" SOC. Is it the carbon content adsorbed on clay minerals? Does it also consider the carbon occluded in microaggregates that are physically stable against dispersion?"*

**Our response:** We thank the reviewer for this comment. The proxy for the stabilized SOC is the amount of mineral-associated carbon adsorbed on clay minerals and pedogenic oxides (C mass of the free silt and clay fraction) and the carbon occluded in the microaggregates (C mass of the microaggregate fraction). Since the amount of unprotected carbon (coarse particulate organic matter fraction) is negligible, this manuscript focuses on the carbon associated with microaggregates and the free silt and clay fraction. Note that in that sense we consider the small amount of particulate organic matter in aggregates also to be protected by minerals. We have now changed the first sentence of hypothesis (ii) for better clarification:

"Chemical variability in parent material will result in contrasting geochemical soil properties that affect the formation of C stabilization mechanisms against microbial decomposition (organo-mineral associations and soil aggregates) and hence govern pattern of SOC stocks and stability as a function of C associated with stable microaggregates and the free silt and clay fraction (mineral-associated C)."

REVIEWER#1 COMMENT 4 *"How many soil profiles were sampled? What is the classification of these soil profiles that you sampled? It needs to be shown in the methods section "2.2 Study design and soil sampling."*

**Our response**: The reviewer is right that this important information is indeed missing in the manuscript. The following paragraph has thus been amended to section "2.2 Study design and soil sampling":

"In total, 36 soil cores were sampled (12 soil cores per geochemical region across four topographic positions in triplicate) on which the soil analysis was conducted. In addition, one soil pit down to at least 100 cm per topographic position was dug in the center in one of the three replicate plots in each region and described according to FAO guidelines (FAO, 2006). The soils were classified after World Reference Base (WRB) soil classification (IUSS WRB, 2015). Soils in the mafic region can be described as umbric, vetic and geric ferralsols and ferralic vetic Nitisols. Soils in the felsic regions are classified as geric and vetic Ferralols. The mixed sedimentary region shows geric and vetic Ferralsols along plateaus and slopes, whereas soils at the valley bottoms are described as fluvic Gleysols."

*Used Literature:*

FAO: Guidelines for soil description, 4th edition, 4th ed., FAO, Rome. Available from: 950 http://www.fao.org/3/a-a0541e.pdf, 2006.

FAO: World references base for soil resources 2014. International soil classification system for 953 naming soils and creating legends for soil maps. Update 2015. Food and Agriculture Organization 954 of the United Nations, Rome, Italy, 203 pp., 2014.

Bonell, M.: Runoff generation in tropical forests, in: Forests, Water and People in the Humid Tropics, edited by: Bonell, M., and Bruijnzeel, L. A., Cambridge University Press, New York, USA, 314-406, 9780521829533, 2005.

Douglas, I., and Guyot, J. L..: Erosion and sediment yield in the humid tropics, in: Forests, Water and People in the Humid Tropics, edited by: Bonell, M., and Bruijnzeel, L. A., Cambridge University Press, New York, USA, 407-421, 9780521829533, 2005.
* * *
*REVIEWER#1 COMMENT 5: "L22: I think you meant "changes in hydraulic conditions.""*

**Our response:** The authors thanks for pointing this out to us. In the context of our study, the term "changes in *hydraulic conditions" is the better* choice. We explained the terms "fluvial and hydraulic conditions" in more detail in the context of our study sites. The sentences have now been corrected and amended accordingly:

"Instead, fluvial dynamics and changes in soil moisture conditions had a secondary control on SOC dynamics in valley positions. Valley bottoms might be affected not only by sediments derived from associated hillslopes but also from material redistribution in the entire catchment during flood events (Douglas and Guyot, 2005). In addition, soil moisture conditions at the valley bottoms might be affected not only from interflow from the hillslopes but also from temporarily high ground water levels (Bonell, 2005). These fluvial and hydraulic conditions lead to higher SOC stocks there than at the non-valley positions. "

*Used Literature:*

Bonell, M.: Runoff generation in tropical forests, in: Forests, Water and People in the Humid Tropics, edited by: Bonell, M., and Bruijnzeel, L. A., Cambridge University Press, New York, USA, 314-406, 9780521829533, 2005.

Douglas, I., and Guyot, J. L..: Erosion and sediment yield in the humid tropics, in: Forests, Water and People in the Humid Tropics, edited by: Bonell, M., and Bruijnzeel, L. A., Cambridge University Press, New York, USA, 407-421, 9780521829533, 2005.
* * *
*REVIEWER#1 COMMENT 6: "L23/24: What is "fossil organic carbon (FOC)"? Is it defined by a specific organic compound (e.g., polycyclic aromatic hydrocarbon) or by a certain "age" (e.g., >10000 years)? The authors should define "fossil organic carbon" when they first mentioned it in the abstract."*

**Our response:** This is an important comment. Fossil organic carbon (FOC) is of geogenic origin. It is organic carbon deposited during sedimentation and undergoes coalification or kerogen transformation during diagenesis (Buseck and Beysacc, 2014). In our study, FOC is characterized by high C / N ratios, depleted in N and free of $^{14}$C due to the high age of rock formation. We have now defined the term FOC more precisely in the according sentence in the abstract:

"At several sites we also detected fossil organic carbon (FOC), which is organic C of geogenic origin and is characterized by high C / N ratios, depletion of N and free of $^{14}$C. Here, FOC constitutes up to $52.0 \pm 13.2$ % of total SOC stock in the C depleted subsoil."

*Used Literature:*

Buseck, P. R. and Beyssac, O.: From organic matter to graphite: Graphitization, Elements, 10, 421–426, https://doi.org/10.2113/gselements.10.6.421, 2014.

*REVIEWER#1 COMMENT 7: "L29: What depth interval are you considering for this observation about SOC stocks? 0-20 cm, 0-50 cm, 0-70 cm, 0-100 cm?"*

**Our response:** The reviewer addresses an important point regarding the missing information about the considered depth intervals. The regression models considered three depth intervals for all dependent variables: 0 - 10 cm (topsoil), 30 - 40 cm (shallow subsoil) and 60 - 70 cm (deep subsoil). These depth intervals were selected as they differ in C input and biogeochemical soil factors. Even though the full soil core down to 100 cm was sampled in 10 cm depth increments, only these three selected depth intervals were analyzed in the laboratory to keep the workload at a feasible level. We added this information in the according sentence in the abstract:

"Regressions models, considering depth intervals of 0 - 10 cm, 30 - 40 cm and 60 - 70 cm, showed that variables affiliated with soil weathering, parent material geochemistry and soil fertility, together with soil depth, explained up to 75 % of the variability of SOC stocks and $\Delta^{14}$C."

*REVIEWER#1 COMMENT 8: "L66: What do you mean for "stabilize C in the soil"? stabilize against decomposition? For how long? 100, 1000, 10000 years?"*

**Our response:** The authors agree that the term "C stabilization" should be used with more precision. The sentence was changed to give more information to the reader in following way:

"The accessibility of C for mineralization, however, is predominantly driven by several interacting mechanisms that can stabilize C in soils against decomposition on a decadal up to a millennial time scale (Trumbore, 2000; Trumbore, 2009)."

*Used Literature:*

Trumbore, S.: Age of soil organic matter and soil respiration: radiocarbon constraints on belowground C dynamics, Ecol. Appl., 10, 399-411, https://doi.org/10.1890/1051-0761(2000)010[0399:AOSOMA]2.0.CO;2, 2000.

Trumbore, S.: Radiocarbon and soil carbon dynamics, Annu. Rev. Earth Planet. Sci., 37, 47-66, 10.1146/annurev.earth.36.031207.124300, 2009.

*REVIEWER#1 COMMENT 9: "L66-68: Please, provide the reasoning for why pyrogenic carbon may remain in the soil for centuries. Does it have to do with the following? The degradation of condensed aromatic carbon compounds is an energy-demanding process. For degradation to occur, the biological capacity for specific degradation pathways must exist in the soil [Baldock et al., 2004]. It is likely that certain microorganisms can produce the (costly) enzymes required for pyrogenic carbon degradation and their presence or absence from soils is thus a crucial control on pyrogenic carbon accumulation. However, images of fungal hyphae encasing charcoal particles in soils [Hockaday et al., 2007] indicate that fungi can very well decompose pyrogenic carbon."*

**Our response:** The authors thank the reviewer for suggesting to be more specific about the controls of pyrogenic carbon persistence. The provided literature already points out that pyrogenic carbon cannot simply be classified as an inert C pool but is in contrast dynamic and interacts with soil organisms and the mineral matrix. We therefore changed the sentence in following way:

"For example, certain C compounds such as pyrogenic C show biochemical resistance since the decomposition of its complex molecular structure is an energy demanding process and microbes will preferentially consume more easily available organic C forms (Czimczik and Masiello, 2007; Knicker, 2011)."

*Used Literature:*

Czimczik, C. I., and Masiello, C. A.: Controls on black carbon storage in soils, Global Biogeochem. Cycles, 21, 1-8, 10.1029/2006GB002798, 2007.

Knicker, H.: Pyrogenic organic matter in soil: its origin and occurrence, its chemistry and survival in soil environments, Quat. Int., 243, 251-263, 10.1016/j.quaint.2011.02.037, 2011.

*REVIEWER#1 COMMENT 10: L69: Are you sure that the word "recalcitrant" is the best one to describe what you meant here? Maybe using terms like "long turnover times" may help you to overcome this issue. I highly recommend the authors to read the following publications for further ideas and suggestions:*

*Mikutta, R., Kleber, M., Torn, M.S. and Jahn, R., 2006. Stabilization of soil organic matter: association with minerals or chemical recalcitrance?. Biogeochemistry, 77(1), pp.25-56.*

*Kleber, M., Nico, P.S., Plante, A., Filley, T., Kramer, M., Swanston, C. and Sollins, P., 2011. Old and stable soil organic matter is not necessarily chemically recalcitrant: implications for modeling concepts and temperature sensitivity. Global Change Biol., 17(2), pp.1097-1107."*

**Our response:** We agree with the reviewer that the term "recalcitrant" in the context with fossil organic carbon is misleading. Our study, together with other recent studies, show that fossil organic carbon is dynamic and decomposable (Bukombe et al., 2021; Hemmingway et al., 2018). Hence, the authors have agreed to use the term "long turnover times" instead:

"Another C fraction that is characterized by long turnover times is fossil organic carbon (FOC) (…)."

*Used Literature:*

Bukombe, B., Fiener, P., Hoyt, A. M., Doetterl, S.: Controls on heterotrophic soil respiration and carbon cycling in geochemically distinct Afrcian tropical forest soils, SOIL DISCUSSIONS, in review, https://doi.org/10.5194/soil-2020-96, 2021.

Hemingway, J. D., Hilton, R. G., Hovius, N., Eglinton, T. I., Haghipour, N., Wacker, L., Chen, M.-C., and Galy, V. V.: Microbial oxidation of lithospheric organic carbon in rapidly eroding tropical mountain soils, Science, 360, 209–212, https://doi.org/10.1126/science.aao6463, 2018.

*REVIEWER#1 COMMENT 11: "L116: Define "geogenic carbon"? Is it the same as "fossil carbon"?"*

**Our response:** The reviewer is right in recognizing this inconsistency. It is meant to be "fossil organic carbon". We have now changed this through the manuscript to the latter.

*REVIEWER#1 COMMENT 12: "L117: You said "more active microbial communities". However, you didn't conclude the comparison. Is it more active than what or where?"*

**Our response:** Thank you for pointing this out. We will conclude the comparison in the following way:

"Similarly, fossil organic carbon that is brought to the surface might become increasingly decomposed when brought in contact with more active and abundant topsoil microbial communities compared to subsoil environments."

*REVIEWER#1 COMMENT 13: "L140: Check the concordance of the preposition "on" and the verb "affect" in this sentence: "our current understanding on how geochemistry and topography in highly weathered tropical soils affects SOC stocks and stabilization mechanisms is still limited."*

**Our response:** Thanks for this comment. The preposition will be corrected in following way:

"In summary, our current understanding of how geochemistry and topography in highly weathered tropical soils affects SOC stocks and stabilization mechanisms against microbial decomposition is still limited."

*REVIEWER#1 COMMENT 14: "L225: What method did you use for "texture" determination? Hydrometer, pipette?"*

**Our response:** We used the Bouyoucos hydrometer method (Bouyoucos, 1962) modified following Beretta et al. (2014). To keep the "2.3 Soil analysis" section streamlined, we decided to focus on the SOC fractionation and pedogenic extraction scheme. A more detailed description of all other soil analysis listed in the manuscript can be found in the project-related database publication (Doetterl et al., 2021a; 2021b). However, if the reviewer thinks that all used methods should be described in detail, we will expand the according section.

*Used literature:*

Beretta, A. N., Silbermann, A. V., Paladino, L., Torres, D., Bassahun, D., Mussell, R., and Garciá-Lamohte, A.: Soil texture analysis using a hydrometer: modification of the Bouyoucos method, Cien. Inv. Agr., 42, 263-271, 10.4067/S0718-16202014000200013, 2014.

Bouyoucos, G. J.: Hydrometer method improved for making particle size analyses of soils, Agronomy Journal, 54, 464-465, 10.2134/agronj1962.00021962005400050028x, 1962.

Doetterl, S., Asifiwe, R.K., Baert, G., Bamba, F., Bauters, M., Bukombe, B., Cadisch, G, Cizungu, L., Cooper, M., Hoyt, A., Kabaske, C., Kalbitz, K., Kidinda, K.L., Maier, A., Mainka, M., Mayrock, J., Muhindo, D., Mujinya, B., Mukotanyi, S.M., Nabahungu, L., Reichenbach, M., Rewald, B., Six, J., Stegmann, A., Summerauer, L., Unseld, R., van Oost, K., Verheyen, K., Vogel, C., Wilken, F., Fiener, P. Organic matter cycling along geochemical, geomorphic and disturbance gradients in forests and cropland of the African Tropics - Project TropSOC DATABASE_v1.0, Earth System Science Data DISCUSSIONS (pre-print), https://doi.org/10.5194/essd-2021-73, 2021a.

Doetterl, S.; Bukombe, B.; Cooper, M.; Kidinda, L.; Muhindo, D.; Reichenbach, M.; Stegmann, A.; Summerauer, L.; 36 Wilken, F.; Fiener, P. TropSOC Database. Version 1.0. GFZ Data Services. https://doi.org/10.5880/fidgeo.2021.009, 37, 2021b.

*REVIEWER#1 COMMENT 15: "L248: Did you subtract the rock fragment to calculate SOC stocks?"*

**Our response:** The samples were free of rock fragments or the amount was negligible (all soils were deeply weathered). We added this information to the section "Soil C fractionation and nutrient analysis":

"Since the content of rock fragments of all samples were negligible, the SOC stock of the bulk soil (SOC$_{bulk}$) was calculated by just multiplying the SOC concentration with the bulk density and thickness of the depth increment (10 cm)."

*REVIEWER#1 COMMENT 16: "L329: Please, check if this statement is necessary: "For non-valley positions, pyrophosphate extractable oxides (0.02 to 1.93 mass%) and oxalate extractable oxides (0.32 to 2.33 mass%) were low compared to DCB extractable oxides". Assuming that DCB extracts most of the oxides (including oxalate and pyrophosphate extractable oxides), then, DCB extractable oxides will always be the highest regardless of the landscape position."*

**Our response:** Thanks for this comment. The statement is still valid, since we used a sequential pedogenic oxide extraction scheme. In the first step, sodium-pyrophosphate was used to extract organically complexed metals. In the second step, ammonium-oxalate-oxalic acid was used on the same sample to extract short-range order oxides and in the last step, DCB was used again on the same sample to extract crystalline oxides. Therefore, the DCB extract will not contain any pyrophosphate or oxalate extractable oxides. We presented the results of all three oxide fractions separately, since we used each of them in the rotated principal component analysis, in which they loaded on different components.

*REVIEWER#1 COMMENT 17: "Figure 3. It would be helpful if we know the soil types. Are they all ferralsols? Or the soils derived from "mafic" parent material are lixisol and nitisol as well? Also, it would be nice to see in another figure and table the average clay content as a function of soil depth in each group: mafic, felsic, mixed."*

**Our response:** Thank you for your suggestion. We have now added the soil types based on our soil profile descriptions for each region in Figure 3. We also followed the advice of the reviewer to plot the clay content against soil depth for each group with an additional table for reviewers and editors information (Figure R1, Table R1). It shows that the clay content increases with soil depth and is highest in mafic soils. To be consistent, we would also add the soil types to Figure 4, Figure S1 and Figure S2 in the same way.

[Figure]

**Figure 3: (a) m / s+c ratio (n = 9 per soil depth) and (b) pedogenic oxide fractions (n = 3 per soil depth) of the sequential extraction across geochemical regions in non-valley positions. Letters indicate significant differences between geochemical regions per soil depth for m / s+c ratio (a) and total pedogenic oxide mass (b). Asterisks indicate no significant differences in means (p > 0.05). Error bar represents standard error.**

[Figure]

**Figure R1: Mean clay content as a function of soil depth across geochemical regions in non-valley positions (n = 9). Error bar represents standard deviation.**

**Table R1: Mean and standard deviation of clay content (%) across geochemical regions in non-valley positions (n = 9).**

| depth increment | mafic (ferralsols, nitisol) | felsic (ferralsols) | mixed (ferralsols) |
|---|---|---|---|
| 0 - 10 cm | 53.51±3.9 | 37.47±4.43 | 37.23±10.95 |
| 30 - 40 cm | 61±4.42 | 41.5±9.95 | 47.66±12.53 |
| 60 - 70 cm | 63.56±6 | 47.75±9.35 | 51.22±14.26 |

*REVIEWER#1 COMMENT 18: "L539: Iron concretions are found, but I am not sure about aluminon concretions. Check whether all these soils contain Fe-concretion or nodules because these concretions are not always found even in microaggregates."*

**Our response:** Thanks for this important comment. We addressed this topic in the discussion as followed:

"Fe-oxides like hematite can incorporate large amounts of Al in their crystal structure by substitution, especially within a kaolinite-rich soil matrix (Tardy and Nahon, 1985). Such (hydro)oxides of Al and Fe act as a cementing agent in the formation of pseuosands and their chemical composition is controlled by parent material geochemistry. In general, (hydro)oxides dominated by Fe are more abundant on mafic rocks, whereas Al is more abundant on rocks with low amounts of Fe (e.g. quartz-rich sedimentary rocks) (Martinez and Souza, 2020). In our study both, the parent material and soil geochemistry show considerable amounts of Al even though the Fe-content exceeds that of Al. The dominating soils are ferralsols and nitisols, which are dominated by kaolinitic clays. Given the above mentioned observations, it is likely that Fe- and Al-concretions are present in the studied soils, even though the elemental composition of the concretions were not directly measured."

*Used Literature:*

Martinez, P., and Souza, I. F.: Genesis of pseudo-sand structure in Oxisols from Brazil – a review, Geoderma Regional, 22, https://doi.org/10.1016/j.geodrs.2020.e00292, 2020.

Tardy, Y., and Nahon, D.: Geochemistry of laterites, stability of Al-goethite, Al-hematite, and Fe3+-kaolinite in bauxites and ferricretes: an approach to the mechanism of concretion formation, Am. J. Sci., 285, 865-903, 1985.
* * *
*REVIEWER#1 COMMENT 19: "L540: The citation Martinez and Souza, 2019 is actually from 2020. Check it in other parts of the manuscript."*

**Our response:** The reviewer is right. Thank you for pointing this out. The manuscript was citing the pre-proof version of this publication. The citation is now updated throughout the manuscript.
* * *
*REVIEWER#1 COMMENT 20: "L543: Check the following publication to obtain more information about the effects of soil depth and texture on carbon retention in Ferralsols. They may help you with new insights and allow comparisons with your data. Soil depth and texture seem to regulate the content of carbon retained by the mineral matrix in Ferralsols. Make sure you addressed it in the discussion.*

*Souza, I.F., Almeida, L.F.J., Jesus, G.L., Kleber, M. and Silva, I.R., 2017. The mechanisms of organic carbon protection and dynamics of C-saturation in Oxisols vary with particle size distribution. European Journal of Soil Science, 68(5), pp.726-739.*

*Souza, I.F., Almeida, L.F., Jesus, G.L., Pett-Ridge, J., Nico, P.S., Kleber, M. and Silva, I.R., 2018. Carbon sink strength of subsurface horizons in Brazilian oxisols. Soil Sci. Soc. Am. J., 82(1), pp.76-86."*

**Our response:** We thank the reviewer for the literature suggestion as it provides further insights in our data. To address the effect of soil depth on C retention we conducted a correlation analysis using the rotated components and SOC stocks separately for each depth increment. The result is comparable to Souza et al. (2018), where C stabilization via interaction with the mineral soil matrix is minor in topsoils but becomes more important in subsoils (Table R2). Please note, that Table R2 is for reviewer and editor information only and we would not include it in the manuscript. We addressed the impact of soil depth on C content in the discussion section in the following way:

"On the contrary, the C stabilization with the geochemical predictors is more effective in shallow and deep subsoils compared to topsoils as indicated by insignificant correlations in the topsoils but significant correlation in subsoils (data not shown). This indicates that the capacity to stabilize C is reached due to high C input in contrast to subsoils receiving less C hence less saturated mineral surfaces (Souza et al., 2018)."

**Table R2: Pearson correlation between SOC$_{bulk}$ and rotated components for each depth increment.**

| Soil depth | | rPC1$_{nv}$ - solid phase | rPC2$_{nv}$ - liquid phase | rPC3$_{nv}$ - silt content | rPC4$_{nv}$ - organic-metal complexes |
|---|---|---|---|---|---|
| 0 - 10 cm | Pearson Correlation | 0.366 | -0.149 | 0.09 | 0.162 |
| | Sig. | 0.332 | 0.701 | 0.818 | 0.678 |
| | N | 9 | 9 | 9 | 9 |
| 30 - 40 cm | Pearson Correlation | 0.681* | -0.927** | -0.791* | 0.931** |
| | Sig. | 0.043 | 0 | 0.011 | 0 |
| | N | 9 | 9 | 9 | 9 |
| 60 - 70 cm | Pearson Correlation | 0.509 | -0.822** | -0.719* | 0.816** |
| | Sig. | 0.162 | 0.007 | 0.029 | 0.007 |
| | N | 9 | 9 | 9 | 9 |

*Correlation is significant at the 0.05*

*** Correlation is significant at the 0.01*

The effect of texture as a function of clay content on the potential to stabilize C (especially in the fraction < 53 µm) as shown by Souza et al. (2017) could not be detected in our data. No significant correlations were found between clay content, SOC$_{bulk}$ or C associated with the free silt and clay fraction (< 53 µm). This can be probably explained by the very different experiment design. In the study of Souza et al. (2017), the particle size distribution was manipulated to investigate the effect of changing soil texture on the C saturation deficit in tropical soils. They argue that the manipulation of the particle size distribution leads to large C losses. Further, reactive pedogenic oxide surfaces became exposed by the experiment treatment, which were previously involved in stabilizing aggregates. This difference in the experimental design may explain why clay is not that important in our study. We therefore would just add the reference and only slightly adjust the corresponding sentence about clay content:

"Clay content, identified as a major factor for stabilizing SOC in temperate soils (Angst et al., 2018) and also in tropical soil systems (Quesada et al., 2020; Souza et al., 2017) was not identified as a major control for our soils."

*Used Literature:*

Angst, G., Messinger, J., Greiner, M., Häusler, W., Hertel, D., Kirfel, K., Kögel-Knabner, I., Leuschner, C., Rethemeyer, J., and Mueller, C. W.: Soil organic carbon stocks in topsoil and subsoil controlled by parent material, carbon input in the rhizosphere, and microbial-derived compounds, Soil Biol. Biochem, 122, 19–30, https://doi.org/10.1016/j.soilbio.2018.03.026, 2018.

Quesada, C. A., Paz, C., Mendoza, E. O., Phillips, O. L., Saiz, G., and Lloyd, J.: Variations in soil chemical and physical properties explain basin-wide Amazon forest soil carbon concentrations, SOIL, 6, 53–88, https://doi.org/10.5194/soil-6-53-2020, 2020.

Souza, I. F., Almeida, L. F. J., Jesus, G. L., Kleber, M., and Silva, I. R.: The mechanisms of organic carbon protection and dynamics of C-saturation in Oxisols vary with particle-size distribution, European Journal of Soil Science, 68, 726-739, 10.1111/ejss.12463, 2017.

Souza, I. F., Almeida, L. F. J., Pett-Ridge, J., Nico, P. S., Kleber, M., and Silva, I. R.: Carbon sink strength of subsurface horizons in Brazilian Oxisols, Soil Sci. Soc. Am. J., 82, 76-86, 10.2136/sssaj2017.05.0143, 2018.

**Errata found in manuscript**

The legend in Figure 5, explaining the meaning of color and shape of the dots, is missing. It will be corrected as follows:

[Figure]

**Figure 5: Biplots of the varimax rotated principal component analysis. (a) rPC1nv and rPC2nv and (b) rPC3nv and rPC4nv of non-valley positions (n = 27). Observations cluster together based on similarities within geochemical regions and their distinction to other geochemical regions. Vector length indicates how strongly variables influence a specific rPC. The angles between vectors display the degree of auto-correlation between variables. Small angles represent positive correlations and high degree of autocorrelation, diverging angles represent negative correlations and high degree of autocorrelation, high angles indicate no correlations between variables and/or rPCs.**

We hope we have addressed all concerns and look forward to hearing from you.

 Best regards,

The authors

---

## Author Comment (AC3)

**Point-by-point response to anonymous Referee#2 comments**

Dear Referee#2,

We would like to thank you for your time and thorough evaluation of our manuscript "*The role of geochemistry in organic carbon stabilization in tropical rainforest soils*", (https://doi.org/10.5194/soil-2020-92). We are very pleased that you positively assessed our work and recognized its relevance. Your comments helped us to significantly improve our manuscript and we want to sincerely thank you for the constructive and valuable insights.

We have addressed all comments and suggestions to the best of our ability. Please find below a point-by-point response to all the concerns raised and how we addressed them. Reviewer original comments are highlighted in grey. New text to be added or modified in the manuscript has quotation marks and is blue-colored in the response.

We hope you find our response and changes to the manuscript satisfying and we are looking forward to hearing from you.

Yours sincerely,

The authors

*REVIEWER#2 COMMENT 1: "However, the most important concern for me is the research design, why the slope plots are 6 while valley and plateau are 3 each?"*

**Our response:** The reviewer points out an important detail of the sample design related to topography of the investigated landscape. We describe in section "2.2 Study design and soil sampling": The dominating landscape feature in all study regions were slopes, whereas plateau and valley positions were much smaller compartments. We also wanted to assess the hypothetical effect of different hillslope positions (different in slope and curvature) on erosion and SOC stocks under tropical rainforests (Dialynas et al., 2016). Hence, it is not possible to describe the soil conditions and its erosional modification among slopes with just one sampled position. We therefore sampled at both topslope and midslope positions to account for their spatial extent and different geometry along the catenae. During pretests, ANOVA showed no significant differences in SOC stock means across top- and midslopes in each study region. As such, we decided to group the two slope positions together in further ANOVAs to reduce redundancy in the topographic position grouping.

*Used Literature:*

Dialynas, Y. G., Bastola S., Bras, R. L., Marin-Spiotta, E., Silver, W. L., Arnone, E., and Noto, L. V.: Impact of hydrologically driven hillslope erosion and landslide occurrence on soil organic carbon dynamics in tropical watersheds, Water Resour. Res., 52, 8895-8919, 10.1002/2016WR018925, 2016.

*REVIEWER#2 COMMENT 2: "How could you justify the absence of soil erosion at the sampling plots as mentioned in your results section while sediment soils were collected in the valley plots? Kindly elaborate on this in your discussion."*

**Our response:** Thanks for this comment. We reported in section "3.1 Climate and topography": We could not find significant differences in the means of SOC stocks and geochemical soil properties between plateau and slope positions. In the case of erosion, we would have expected some differences in soil properties and SOC stocks between these positions. The only differences were caused by valley positions because of fluvial activities unrelated to hillslope processes along the investigated catenae.

This was confirmed by another study analyzing $^{239+240}$Pu activity and inventories as a means for direct measurement of erosional soil removal. Here we found that the $^{239+240}$Pu inventories, sampled along the same catenae as used in our study did not show topographic patterns, which indicates little or no soil erosion (Wilken et al., 2020). We therefore excluded valley positions from further analysis in the manuscript and kindly refer to the supplementary results and short discussion therein.

*Used literature:*

Wilken, F., Fiener, P., Ketterer, M., Meusburger, K., Muhindo, D. I., van Oost, K., and Doetterl, S.: Assessing soil erosion of forest and cropland sites in wet tropical Africa using $^{239+240}$Pu fallout radionuclides, SOIL DISCUSSIONS, in review, https://doi.org/10.5194/soil-2020-95, 2020.

*REVIEWER#2 COMMENT 3: "Though in forest, there is no or little surface soil erosion but landslides occur most of the time in Tropical forest."*

**Our response:** This is an important comment. During fieldwork, we could indeed observe landslides after heavy rainfalls on very exposed and steep slopes along roadcuts. However, those areas were free from vegetation and usually strongly altered by human activity. We were aware of this during our scouting trips and paid attention to install the study plots in areas that are as little as possible affected by landslides to exclude the effect of the latter thus focusing on the hypothesized effect of surface soil erosion which takes place at the broader hillslope-scale. While we cannot exclude our areas to be affected by naturally occurring landslides, we can exclude those events for the time needed to establish the current vegetation coverage since vegetation patterns were fairly regular across landforms and replicates. Additionally, landforms and sampled soils did not show signs of larger erosional events in the recent past. All soils were deeply weathered and showed (outside of valleys and fluvial systems) no layering or other signs that would indicate a disturbance event in the past. We added this information to section "2.2 Study design and soil sampling":

"Attention was paid to install the study plots in areas that are as little as possible affected by landslides. The occurrence of natural landslides cannot be excluded with certainty. However, the vegetation patterns were fairly regular across landforms and replicates thus events can be excluded for the time needed to establish the current vegetation coverages. Additionally, landforms and sampled soils did not show signs of larger erosional events in the recent past. All soils were deeply weathered and showed no layering or other signs that would indicate a disturbance event in the past outside of valleys and fluvial systems. "

*REVIEWER#2 COMMENT 4: "I understand that one of the main focuses was Geochemistry of the three sites, while you also recorded soil fertility parameters especially chemical. but anywhere biological parameter such enzymes related to carbon cycle such as b-glucosidase which decompose carbon source is important for the decomposition were recorded? or mentioned in relation to the study."*

**Our response:** The authors thank the reviewer for this very important comment. Soil microbial related properties are crucial to understand SOC dynamics in a given setting. As such, microbial biomass carbon and extracellular enzyme activity were measured and explored in more detail by our colleagues during a 120-day incubation experiment on the same soil samples used in our study, the results of which are being prepared for a separate manuscript (Kidinda et al., 2021). As the reviewer already stated, the focus of our manuscript was the impact of geochemistry on SOC dynamics in contrasting geochemical regions under tropical conditions. Therefore, we kindly refer to the connected work of Kidinda et al. (2021) published in the same special issue for details on the microbial activity along the investigated soil sequences.

*Used literature:*

Kidinda, L. K., Olagoke, F. K., Vogel, C., Kalbitz, K., and Doetterl, S.: Patterns of microbial processes shaped by parent material and soil depth in tropical rainforest soils, SOIL DISCUSSIONS, in review, https://doi.org/10.5194/soil-2020-80, 2020.

*REVIEWER#2 COMMENT 5: "1) could you elaborate why the slope of you compared sites of different slopes eg., Kibale site (3-55%) while Kahuzi-Biega and Nyungwe have similar slopes (1-60%)."*

**Our response:** Thanks for this comment. We tried our best to keep the slopes comparable across the study regions, but this was not always possible. However, there are no significant differences in the means of the slopes across study regions ($p = 0.97$) and no significant correlations between slope and SOC stock ($p = 0.63$). Therefore, we are confident that the differences in the slope range between study regions were not affecting our target variables.

*REVIEWER#2 COMMENT 6: "2) Soil development is dependent to several factors including environment, which are different from the three. What was your reference of standardization to justify the study comparison between the sites."*

**Our response:** Thanks for this important question. Our choice for the study sites was based on prior knowledge of similarities between vegetation structure, climate and topography. Overall, the soil forming factors are comparable among our study sites except for parent material, which differed in geochemistry and texture (Figure R1). As different parent materials impact geochemical soil properties and therefore the potential to stabilize SOC, we hypothesized that parent material geochemistry would shape patterns of SOC stocks and soil C fractions the most.

[Figure]

**Figure R1: Chemical composition of unweathered rock samples representing the parent material for soil formation in three studied geochemical regions (mean +/- standard error). Panel 3a shows the distribution and concentration of rock-derived aluminum (Al), iron (Fe) and manganese (Mn) and total silica content (Si). Panel 3b shows the distribution and concentration of rock-derived calcium (Ca), potassium (K), magnesium (Mg), sodium (Na) and phosphorus (P). Note the difference in scale on y-axis between panel 3a and 3b (Doetterl et al., 2021a; 2021b).**

*Used Literature:*

Doetterl, S., Asifiwe, R.K., Baert, G., Bamba, F., Bauters, M., Bukombe, B., Cadisch, G, Cizungu, L., Cooper, M., Hoyt, A., Kabaske, C., Kalbitz, K., Kidinda, K.L., Maier, A., Mainka, M., Mayrock, J., Muhindo, D., Mujinya, B., Mukotanyi, S.M., Nabahungu, L., Reichenbach, M., Rewald, B., Six, J., Stegmann, A., Summerauer, L., Unseld, R., van Oost, K., Verheyen, K., Vogel, C., Wilken, F., Fiener, P. Organic matter cycling along geochemical, geomorphic and disturbance gradients in forests and cropland of the African Tropics - Project TropSOC DATABASE_v1.0, Earth System Science Data DISCUSSIONS (pre-print), https://doi.org/10.5194/essd-2021-73, 2021a.

Doetterl, S.; Bukombe, B.; Cooper, M.; Kidinda, L.; Muhindo, D.; Reichenbach, M.; Stegmann, A.; Summerauer, L.; 36 Wilken, F.; Fiener, P. TropSOC Database. Version 1.0. GFZ Data Services. https://doi.org/10.5880/fidgeo.2021.009, 37, 2021b.

*REVIEWER#2 COMMENT 7: "3) in material and methods you are referring to Fick and Hijmans, 2017. is his work conducted in all your study regions?"*

**Our response:** This is indeed an important question. The outcome of Fick and Hijams (2017) is the WorldClim 2 dataset, which provides spatially interpolated monthly climate data for global land areas at a spatial resolution of approximately 1 km² using data from between 9,000 to 60,000 weather stations with a temporal range of 1970-2000. Datasets used for representing covariates and climate elements in the tropics are from the Center for Tropical Agriculture (CIAT) in Columbia. The WorldClim 2 dataset is suitable for comparing the different regions but not for comparing the climatic variability along the investigated catenae within the regions due to the coarse resolution of this global dataset. We installed three weather stations (ATMOS 41, Meter, Germany) in each geochemical region close to the investigated catenae. This enabled us to collect micrometeorological data at a temporal resolution of 5 minutes on precipitation, air temperature, relative humidity and air pressure (Doetter et al., 2021a; 2021b). However, these local climate stations only recorded for 2.5 years by now which is why we resort to the larger scale but coarser WorldClim 2 dataset.

*Used literature:*

Doetterl, S., Asifiwe, R.K., Baert, G., Bamba, F., Bauters, M., Bukombe, B., Cadisch, G, Cizungu, L., Cooper, M., Hoyt, A., Kabaske, C., Kalbitz, K., Kidinda, K.L., Maier, A., Mainka, M., Mayrock, J., Muhindo, D., Mujinya, B., Mukotanyi, S.M., Nabahungu, L., Reichenbach, M., Rewald, B., Six, J., Stegmann, A., Summerauer, L., Unseld, R., van Oost, K., Verheyen, K., Vogel, C., Wilken, F., Fiener, P. Organic matter cycling along geochemical, geomorphic and disturbance gradients in forests and cropland of the African Tropics - Project TropSOC DATABASE_v1.0, Earth System Science Data DISCUSSIONS (pre-print), https://doi.org/10.5194/essd-2021-73, 2021a.

Doetterl, S.; Bukombe, B.; Cooper, M.; Kidinda, L.; Muhindo, D.; Reichenbach, M.; Stegmann, A.; Summerauer, L.; 36 Wilken, F.; Fiener, P. TropSOC Database. Version 1.0. GFZ Data Services. https://doi.org/10.5880/fidgeo.2021.009, 37, 2021b.

Fick, S. E., and Hijmans, R. J.: WorldClim 2: new 1-km spatial resolution climate surfaces for global land areas, Int. J. Climatol., 37, 4302–4315, https://doi.org/10.1002/joc.5086, 2017.

*REVIEWER#2 COMMENT 8: "4) How could you compared results from three different slope length beside variability sites elevation. e.g., Nyungwe and Kibale sites are more variable than Kahuzi biega."*

**Our response:** The authors like to thank the referee for this question, which helped us to recognize a mistake in our slope length calculation. We corrected the paragraph as follows:

"Slope length in Kahuzi-Biéga was 70±56 m (max. 170 m), in Nyungwe 101±103 m (max. 339 m) and in Kibale 149±125 m (max. 374 m)."

The slope length shows indeed high variances across the study regions with different maximum slope lengths. But there are no significant correlations between SOC stocks and slope length. Regarding the minor or absent soil erosion in our study sites, we considered slope length as an irrelevant factor in explaining SOC stocks at the plot scale under pristine tropical rainforest. Even though shallow subsoil SOC stocks (30-40 cm) show significant correlations with altitude ($p < 0.01$), they become non significant when controlled for geochemical soil properties (DCB extr. oxides, exchangeable bases, total P). This orographic effect as a function of MAP and MAT is interpreted as a second-order control which affects SOC stocks indirectly via geochemical soil properties. As such, our study site comparison regarding SOC stocks should not be biased by the variability in elevation and slope length. However, we admit that the more complex topography in our study regions ask for the catchment size where the plots are located instead of just using the slope length as a proxy for soil erosion. Due to the weak DEM this could not be calculated in the required precision.

*REVIEWER#2 COMMENT 9: "5) How can you explain the absence of soil erosion in plots, while the sampling was conducted during rainy season (March to June)?"*

**Our response:** The reviewer points out a very important remark, since the tropical rainforest climatic type shows the highest rainfall erosivity (Panagos et al., 2017). At the same time, global erosion studies from tropical forest sites show rather low mean erosion rates of 0.2 Mg ha-1 yr-1 compared to other climate zones (Xiong et al., 2019). This can be attributed to a variety of interactions between precipitation, standing vegetation and organic soil layers. For example, closed canopy covers, understoreys, litter and organic soil layers reduces the kinetic energy of raindrops significantly thus decreasing splash erosion and therefore preventing i.a. soil crusting which in turn affects the soil infiltration capacity (Labriére et al., 2015; Singer and Shainberg, 2004). The litter layer and ground vegetation helps to prevent soil erosion by funneled stemflow (Dunkerley, 2020). But also plant roots enhance soil erosion resistance (Li et al., 2017). Our study sites showed all above mentioned features of multiple layers of vegetation, organic soil layers and roots, which prevents soil erosion as a result of heavy rainfall events during the rainy season.

*Used Literature:*

Dunkerley, D.: A review of the effect of throughfall and stemflow on soil properties and soil erosion, in: Precipitation Partitioning by Vegetation. A Global Synthesis, edited by: van Stan, J. T. I., Gutmann, E., and Friesen, J., Cham, Switzerland, Springer, 183-214, https://doi.org/10.1007/978-3-030-29702-2, 2020.

Labriére, N., Locatelli, B., Laumonier, Y., Freycon, V., and Bernoux, M.: Soil erosion in the humid tropics: A systematic quantitative review, Agric. Ecosyst. Environ. Agriculture, 203, 127-139, https://doi.org/10.1016/j.agee.2015.01.027, 2015.

Li, Q., Liu, G.-B., Zhang, Z., Tuo, D.-G., Bai, R.-R., and Qiao, F.-F.: Relative contribution of root physical enlacing and biochemistrical exudates to soil erosion resistance in the Loess soils, Catena, 153, 61-65, https://doi.org/10.1016/j.catena.2017.01.037, 2017.

Panagos, P., Borrelli, P., Meusburger, K., Yu, B., Klik, A., Lim, K. J., Yang, J. E., Ni, J., Miao, C., Chattopadhyay, N., Sadeghi, S. H., Hazbavi, Z., Zabihi, M., Larionov, G. A., Krasnov, S. F., Gorobets, A. V., Levi, Y., Erpul, G., Birkel, C., Hoyos, N., Naipal, V., Oliveira, P. T. S., Bonilla, C. A., Meddi, M., Nel, W.,Dashti, H. A., Boni, M., Diodato, N., van Oost, K., Nearing, M., and Ballabio, C.: Global rainfall erosivity assessment based on high-temporal resolution rainfall record, Scientific reports, 7, 1-12, 10.1038/s41598-017-04282-8, 2017.

Singer, M. J., and Shainberg, I.: Mineral soil surface crusts and wind and water erosion, Earth Surf. Process. Landdforms, 29, 1065-1075, 10.1002/esp.1102, 2004.

Xiong, M., Sun, R., and Chen, L.: A global comparison of soil erosion associated with land use and climate type, Geoderma, 343, 31-39, https://doi.org/10.1016/j.geoderma.2019.02.013, 2019.

*REVIEWER#2 COMMENT 10: "6) on the paragraph 340 "Not significant correlation was found with the included climate variables (data not shown); For the reader to realize that they were not significant difference a figure or table is require. can you add that?"*

**Our response:** The authors like to thank the referee for pointing this out and we agree to provide more details. Significant correlations between SOC stocks and climatic parameters (MAP, MAT and PET) are only found in the shallow subsoils (30-40 cm), whereas any correlations in topsoils (0-10 cm) and deep subsoils (60-70 cm) are absent. The correlation in the shallow subsoils disappears when controlled for soil properties (see Table R1). As such, SOC stocks are only indirectly affected by climate by its impact on geochemical soil properties and thus do not have independent explanation power. This is our rationale to focus on the direct effect of soil properties on SOC dynamics in our study sites. We will add Table R1 to the manuscript appendix.

However, we have to point out that the global WorldClim 2 dataset is only suitable to compare climatic differences across our study regions but cannot resolve the local variability between plots within the regions. As already mentioned in the response to Reviewer#2 comment 7, we installed weatherstation near the study plots to cover local climatic variability. But since we only have records of 2.5 years so far, we resort to the WorldClim 2 dataset.

**Table R1: Partial correlation analysis between $SOC_{bulk}$ and climate variables (Fick and Hijams, 2017) controlling for geochemical soil properties. Zero-order correlation displays the Pearson r when including no control variables. The controlled correlation shows the Pearson r when controlling for DCB extractable oxides of Al, Fe and Mn, exchangeable bases and total P. *p<0.05; **p<0.001.**

| soil depth [cm] | control variables | MAP | MAT | PET |
|---|---|---|---|---|
| 0-10 | zer-order | -0.17 | 0.00 | -0.08 |
| | DCB extr. oxides (Al, Fe, Mn), exchangeable bases, total P | -0.16 | -0.26 | -0.40 |
| 30-40 | zer-order | 0.67* | -0.90** | -0.93** |
| | DCB extr. oxides (Al, Fe, Mn), exchangeable bases, total P | -0.56 | -0.03 | -0.43 |
| 60-70 | zer-order | -0.06 | -0.24 | -0.33 |
| | DCB extr. oxides (Al, Fe, Mn), exchangeable bases, total P | 0.00 | -0.13 | -0.14 |

*Used Literature:*

Fick, S. E., and Hijmans, R. J.: WorldClim 2: new 1-km spatial resolution climate surfaces for global land areas, Int. J. Climatol., 37, 4302–4315, https://doi.org/10.1002/joc.5086, 2017.

*REVIEWER#2 COMMENT 11: "7) on 360; You mentioned that total P was high in Mafic region as compared to both mixed sedimentary rocks and felsic regions. How could you justify your finding with the known situation of low available P in that region?"*

**Our response:** This is an important question. Our rationale is the specific mineralogy and mafic geochemistry of alkaline basaltic rocks. Basalts consist of primary minerals like olivin, pyroxene and Ca-rich feldspars which contain P in their crystal structure. Compared to acid plutonics (i.e. granites) and mixed sedimentary rocks, basalt can contain up to 2.5 times more P. P-release from basalts into the soil matrix by chemical weathering is particularly high in humid areas with high temperatures like in our study sites (Hartmann et al., 2014). Furthermore, the content of bio-available P in highly weathered soil is increased when amended with basaltic material (Gillman et al., 2002) which again underlines the importance of basalts as a P-source for soils. This is further illustrated when comparing P content in bedrocks and soils. Bedrock geochemistry produces a strong difference in total P in unweathered rock samples (Figure R1) which is mirrored in the total P content in soils albeit to a lesser degree (Figure R2). This consolidates our interpretation, that parent material geochemistry leaves a footprint in soil geochemistry besides prolonged chemical weathering in the investigated soils. However, the amount of bio available P seems not solely dependent on bedrock geochemistry as shown by the similar content in bio available P between the mafic and felsic region as well by the much higher P fraction ratio in the latter (Figure R3, for referee and editor information only).

[Figure]

**Figure R2: P fractions in the shallow subsoil (30 - 40 cm) of non-valley positions across geochemical regions. Left: total P; Right: bio available P; Right: Ratio of bio available P and total P. Bar represents mean and standard error shows standard deviation. Per bar n = 3.**

*Used Literature:*

Gillman, G. P., Burkett, D. C., and Coventry, R. J.: Amending highly weathered soils with finely ground basalt rock, Appl. Geochem., 17, 987-1001, https://doi.org/10.1016/S0883-2927(02)00078-1, 2002.

Hartmann, J., Mossdorf, N., Lauerwald, R., Hinderer, M., and West, A. J.: Global chemical weathering and associated P-release - The role of lithology, temperature and soil properties, Chem. Geol., 363, 145-163, http://dx.doi.org/10.1016/j.chemgeo.2013.10.025, 2014.

*REVIEWER#2 COMMENT 12: "8) At 365; How the reader could know that the study region is highly weathered without displaying soil data of all the three depths?"*

**Our response:** Thanks for this comment. We described the soil weathering stage by briefly presenting the chemical alteration index (CIA) in one sentence in the section "Parent material geochemistry and weathering stage" and by presenting the nutrient depletion as a result of weathering in greater detail both in the same section and in the appendix. Figure R3 shows the ratio of $Fe_{dcb}$ versus $Fe_{total}$. This ratio is high in all regions and depth increments which reflects the highly advanced soil weathering stage. This corresponds with the pronounced reddish soil color and absence of rock fragments. To reduce redundancy, we would leave Figure R3 for reviewer and editor information only. However, if the referee and editors share the opinion it would enhance the clarity of the results, we are happy to add Figure R3 to the appendix.

[Figure]

**Figure R3: Fe_dcb / Fe_total ratio against soil depth for each geochemical region for non-valley positions. Datapoints represent mean and standard errors show standard deviation. For each data point n = 3.**

REVIEWER#2 COMMENT 13: *"9) at 375; How could you explain the high depletion of P (72 to 14 %) in parent material under natural conditions? (without any agricultural activity that could contribute to P removal)"*

**Our response:** Thanks for this interesting question. The high P depletion in deeply leached tropical soils in the absence of geological (i.a. volcanism, tectonic uplift) and anthropogenic disturbances (i.a. soil erosion, fertilization) is best explained by progressive loss of P during long term soil development (Vitousek et al., 2010; Walker and Syers, 1976), but also via seasonally driven P leaching at the beginning of the rainy season (Campo et al., 1998). In addition, the P released from primary minerals into the soil matrix via weathering, can accumulate in biologically-available pools like litter and organic soil layers (Silver, 1994; Vitousek et al., 2010). These pools represent a sink, since P will be withdrawn from the mineral soil matrix by plant uptake and recycled between organic layers and plants (Vitousek et al., 2010; Wilcke et al., 2002).

*Used Literature:*

Campo, J., Jaramillo, C. J., and Maass, J. M.: Pulses of soil phosphorus availability in a Mexican tropical dry forest: effects of seasonality and level of wetting, Oecologia, 115, 167-172, 1998.

Silver, W. L.: Is nutrient availability related to plant nutrient use in humid tropical forests? Oecologia, 98, 336-343, 1994.

Vitousek, P. M., Porder, S., Houlton, B. Z. and Chadwick, O. A.: Terrestrial phosphorus limitation: mechanisms, implications, and nitrogen-phosphorus interactions, Ecol. Appl., 20, 5-15, https://doi.org/10.1890/08-0127.1, 2010.

Walker, T. W., and Syers, J. K.: The fate of phosphorus during pedogenesis, Geoderma, 15, 1-19, https://doi.org/10.1016/0016-7061(76)90066-5, 1976.

Wilcke, W., Yasin, S., Abramowski, U., Valarezo, C., and Zech, W.: Nutrient storage and turnover in organic layers under tropical montane rain forest in Ecuador, European Journal of Soil Science, 53, 15-2, https://doi.org/10.1046/j.1365-2389.2002.00411.x, 2002.

We hope we have addressed all concerns and look forward to hearing from you.

 Best regards,

The authors

---

## Author Comment (AC4)

**Point-by-point response to anonymous Referee#3 comments**

Dear Referee#3,

We would like to thank you for your time and thorough evaluation of our manuscript "*The role of geochemistry in organic carbon stabilization in tropical rainforest soils*", (https://doi.org/10.5194/soil-2020-92). We are very pleased that you positively assessed our work and recognized its relevance. Your comments helped us to significantly improve our manuscript and we want to sincerely thank you for the constructive and valuable insights.

We have addressed all comments and suggestions to the best of our ability. Please find below a point-by-point response to all the concerns raised and how we addressed them. Reviewer original comments are highlighted in grey. New text to be added or modified in the manuscript has quotation marks and is blue-colored in the response.

We hope you find our response and changes to the manuscript satisfying and we are looking forward to hearing from you.

Yours sincerely,

The authors

*REVIEWER#3 COMMENT 1: "Lines 22-23. I could not find strong evidence to support the claim that "fluvial dynamics and changed hydrological conditions had a secondary control on SOC dynamics in valley positions, leading to higher SOC stocks there than at the non-valley positions". This should be better explained. How can the reader agree that "fluvial dynamics and changed hydrological conditions" can be inferred from the results reported and help to explain higher SOC stocks in valleys than in non-valley positions?"*

**Our response:** Thank you very much for pointing this out. Indeed, we agree that this statement needs more clarification. We can´t provide solid quantitative data to show that soil moisture and oxygen conditions are different in the valley compared to non-valley positions. But based on our qualitative field observations in the valley positions (e.g. nearby river systems, high water saturation) and soil profile descriptions (e.g. fluvial materials, gleyic properties) we concluded that the soil environmental conditions are markedly different compared to non-valley positions. This is especially true for the valley positions in the mixed sedimentary region. Despite having the lowest content of pedogenic oxides, which are highly relevant in C stabilization at the corresponding non-valley positions, we observed similar SOC stocks in the valley as for non-valley positions. Valley positions in the mafic region show even higher SOC stocks although having less pedognic oxides compared to non-valley positions. Based on this missing link between SOC stocks and stabilization partners, we infer that C stabilization in valley positions is dominantly driven by reduced decomposition rates as a function of restricting soil environmental conditions resulting from fluvial activity.

We would expand the short discussion in the supplements to provide further information to the reader.

Additions in supplementary short discussion:

"Valley bottoms might be affected not only by sediments derived from associated hillslopes but also from material redistribution in the entire catchment during flood events (Douglas and Guyot, 2005). In addition, soil moisture conditions at the valley bottoms might be affected not only from interflow from the hillslopes but also from temporarily high ground water levels (Bonell, 2005). These fluvial and hydraulic conditions then lead to higher SOC stocks there than at the non-valley positions by reducing the decomposition rates as a function of restricting soil environmental conditions (Wiaux et al., 2014). For example, despite having the lowest content of pedogenic oxides, which are important stabilization

partners at the non-valley positions, we observe similar SOC stocks in the valley positions as for non-valley positions in the mixed sedimentary region. Valley positions in the mafic region show even higher SOC stocks although having less pedognic oxides compared to non-valley positions."

*Used Literature:*

Bonell, M.: Runoff generation in tropical forests, in: Forests, Water and People in the Humid Tropics, edited by: Bonell, M., and Bruijnzeel, L. A., Cambridge University Press, New York, USA, 314-406, 9780521829533, 2005.

Douglas, I., and Guyot, J. L..: Erosion and sediment yield in the humid tropics, in: Forests, Water and People in the Humid Tropics, edited by: Bonell, M., and Bruijnzeel, L. A., Cambridge University Press, New York, USA, 407-421, 9780521829533, 2005.

Wiaux, F., Cornelis, J.-T., Cao, W., Vanclooster, M., and Van Oost, K.: Combined effect of geomorphic and pedogenic processes on the distribution of soil organic carbon quality along an eroding hillslope on loess soil, Geoderma, 216, 36-47, 10.1016/j.geoderma.2013.10.013, 2014.

*REVIEWER#3 COMMENT 2: "Lines 23-24. I believe the term "Fossil organic carbon" could be more precisely described and referred to as "geogenic organic carbon" in the whole manuscript. In fact, geogenic organic carbon was used by the authors themselves elsewhere in their text (e.g., page 3, line 116)."*

**Our response:** Thanks for pointing out this discrepancy. In fact, this was also a comment from reviewer#1 and I kindly repeat the according response here:

"Fossil organic carbon (FOC) is of geogenic origin. It is organic carbon deposited during sedimentation and undergoes coalification or kerogen transformation during diagenesis (Buseck and Beysacc, 2014). In our study, FOC is characterized by high C / N ratios, depleted in N and free of $^{14}$C due to the high age of rock formation. We have now defined the term FOC more precisely in the according sentence in the abstract:

"At several sites we also detected fossil organic carbon (FOC), which is organic C of geogenic origin and is characterized by high C / N ratios, depletion of N and free of $^{14}$C. Here, FOC constitutes up to $52.0 \pm 13.2$ % of total SOC stock in the C depleted subsoil.""

The term fossil organic carbon describes its origin the best since it is part of the fossil-fuel formation process (Berner, 2003). Therefore, we would like to keep the term "fossil organic carbon (FOC)" throughout the manuscript, since this term is used by other colleagues studying FOC also in a soil biogeochemical context (Bukombe et al., 2021). This helps to be consistent with terms across connected manuscripts in the same special issue.

*Used Literature:*

Berner, R. A.; The long-term carbon cycle, fossil fuels and atmospheric composition, Nature, 426, 323-326, https://doi.org/10.1038/nature02131, 2003.

Bukombe, B., Fiener, P., Hoyt, A. M., Doetterl, S.: Controls on heterotrophic soil respiration and carbon cycling in geochemically distinct Afrcian tropical forest soils, SOIL DISCUSSIONS, in review, https://doi.org/10.5194/soil-2020-96, 2021.

Buseck, P. R. and Beyssac, O.: From organic matter to graphite: Graphitization, Elements, 10, 421–426, https://doi.org/10.2113/gselements.10.6.421, 2014.

*REVIEWER#3 COMMENT 3: "Lines 69-70. To what extent geogenic carbon (FOC) is preserved owing to its inherent chemical properties (e.g., recalcitrance) or the specific conditions (e.g., burial of organic carbon mixed with mineral particles) under sedimentary environments?"*

**Our response:** Thank you very much for this interesting question. FOC is better described as a C fraction with long turnover times and not as a recalcitrant C pool. Our study in accordance with other recent studies show that FOC is dynamic and decomposable (Bukombe et al., 2021; Hemmingway et al., 2018). The decomposition of FOC is most likely a energy demanding process since it undergoes coalification and kerogen transformation (Buseck and Beysacc, 2014) and microbes will preferentially consume more easily available organic C forms similar to the case of pyrogenic C (Czimiczik and Masiello, 2007; Knicker, 2011). The long turnover times are potentially even more enhanced, when the FOC bearing parent material is distal from the surface and not exposed to weathering and microbial processes.

*Used Literature:*

Bukombe, B., Fiener, P., Hoyt, A. M., Doetterl, S.: Controls on heterotrophic soil respiration and carbon cycling in geochemically distinct Afrcian tropical forest soils, SOIL DISCUSSIONS, in review, https://doi.org/10.5194/soil-2020-96, 2021.

Buseck, P. R. and Beyssac, O.: From organic matter to graphite: Graphitization, Elements, 10, 421–426, https://doi.org/10.2113/gselements.10.6.421, 2014.

Czimczik, C. I., and Masiello, C. A.: Controls on black carbon storage in soils, Global Biogeochem. Cycles, 21, 1-8, 10.1029/2006GB002798, 2007.

Hemingway, J. D., Hilton, R. G., Hovius, N., Eglinton, T. I., Haghipour, N., Wacker, L., Chen, M.-C., and Galy, V. V.: Microbial oxidation of lithospheric organic carbon in rapidly eroding tropical mountain soils, Science, 360, 209–212, https://doi.org/10.1126/science.aao6463, 2018.

Knicker, H.: Pyrogenic organic matter in soil: its origin and occurrence, its chemistry and survival in soil environments, Quat. Int., 243, 251-263, 10.1016/j.quaint.2011.02.037, 2011.

*REVIEWER#3 COMMENT 4: "Line 86. Please provide a reference in which the authors have reported pedogenic oxides contents above 50% in the clay fraction of tropical soils. I agree that some tropical soils can exhibit more than 50% of pedogenic oxides, but such soils are not the norm as implied in the text. Please check."*

**Our response:** Thanks for pointing this out as it clearly shows an apparent typo in the manuscript. According to the meta-analysis of Ito and Wagai (2017), tropical soils contain up to 15 % Fe-oxides (and not 50 %) in the clay fraction which is still much higher compared to temperate soils. The sentence will be corrected accordingly:

"Clay-sized mineral fractions in tropical soils are composed of up to 15 % pedogenic oxides, which is usually much higher than in temperate soils (Ito and Wagai, 2017)."

*Used Literature:*

Ito, A., and Wagai, R.: Data descriptor: global distribution of clay-size minerals on land surface for biogeochemical and climatological studies, Sci. Data, 4, 1-11, 10.1038/sdata.2017.103, 2017.

*REVIEWER#3 COMMENT 5: "Line 140. "composition" or "decomposition"?"*

*Our response:* Thanks. This typo will be corrected:

"In contrast, valley positions will show higher SOC stocks compared to plateau positions due to deposition of C-rich topsoil material and limited decomposition of SOC."

*REVIEWER#3 COMMENT 6: "All hypotheses proposed should be rephrased and put into simpler functional relationships e.g., y = f(x). This would reduce verbosity and give the reader a glimpse on how each hypothesis would be effectively tested. I believe the hypotheses can be better used to guide the reader through the Discussion section as well. Hypothesis (i): what parameters would be used/measured to determine the control of topography on lateral fluxes of water and mineral mass? This is not clear to me. Hypothesis (ii): I understood the context, but verbosity can be reduced. Hypothesis (iii): I found the third hypothesis particularly confusing as it includes a reference to "priming effects", which were not measured in this study. Example to rephrase the third hypothesis: iii) Geogenic soil carbon stocks vary more consistently as a function of soil depth than landscape position or soil parent material."*

**Our response:** This is a very important comment. For clarifying hypothesis (i): Using rock-derived macronutrients like P, Ca, Mg and K and C stabilization-relevant mineral phases like pedogenic oxides together with clay content as a proxy for soil redistribution, we would have expected pronounced differences between plateau, slopes and valley positions. In the case of simple soil redistribution along the catenae, we would have expected lower nutrient/ mineral contents at eroding slopes and higher contents at the depositional valley positions (Wiaux et al., 2014). In the case of enhanced weathering along eroding slopes triggered by soil rejuvenation by exposing unweathered parent material to the surface, we would have expected more nutrients/ minerals along the eroding slopes compared to stable plateau positions (Chadwick and Asner, 2016). These topography-driven differences in mineral content should be correlated with differences in SOC stock across the catenae. We would like to refer to the manuscript section "1.3 Topographic controls on SOC dynamics in tropical forests" where the above mentioned processes are briefly discussed. However, we could not observe such differences caused by hillslope processes when using soil geochemical properties as a proxy parameter. This is in line with a recent study of our colleague who analyzed $^{239+240}$Pu activity and inventories as a means for direct measurement of erosional soil removal. Here we found that the $^{239+240}$Pu, inventories, sampled along the same catenae as used in our study did not show topographic patterns, which indicates little or no soil erosion (Wilken et al., 2020).

For clarifying hypothesis (iii): We would like to refer to the connected study of our colleague, who studied the soil microbial activity on the same soil samples. Here, we show that microbial activity and readily available nutrients are highest in the topsoils across all studied gradients. Nevertheless, the authors agree that mentioning priming effects in the hypothesis lead to confusion since it was not directly measured in this study.

Therefore, we would rephrase the hypotheses in the study aim section accordingly to reduce verbosity:

"(i) SOC stocks and geochemical soil properties sensitive to soil redistribution will vary as a function of topographic position.

(ii) C stabilization mechanisms in highly weathered tropical soils will be driven by geochemical soil properties as a function of parent material composition.

(iii) Fossil organic carbon content in C-bearing parent material will vary as a function of soil depth."

*Used Literature:*

Chadwick, K. D., and Asner, G. P.: Tropical soil nutrient distributions determined by biotic and hillslope processes, Biogeochemistry, 127, 273-289, 10.1007/s10533-015-0179-z, 2016.

Kidinda, L. K., Olagoke, F. K., Vogel, C., Kalbitz, K., and Doetterl, S.: Patterns of microbial processes shaped by parent material and soil depth in tropical rainforest soils, SOIL DISCUSSIONS, in review, https://doi.org/10.5194/soil-2020-80, 2020.

Wiaux, F., Cornelis, J.-T., Cao, W., Vanclooster, M., and Van Oost, K.: Combined effect of geomorphic and pedogenic processes on the distribution of soil organic carbon quality along an eroding hillslope on loess soil, Geoderma, 216, 36-47, https://doi.org/10.1016/j.geoderma.2013.10.013, 2014.

*REVIEWER#3 COMMENT 7: "I believe the supplement 1 could benefit the reader if kept in the main text, all results reported therein are very nice. Besides, as far as I understood hypothesis (i), the observation of higher soil C stocks in valleys than in non-valley positions is important for this research."*

**Our response:** Thank you for this positive feedback. We have been thinking about this ourselves and since we wanted to keep the result and discussion part streamlined and focused on mineral-related C stabilization, we would prefer to keep the supplements separate from the manuscript as it deals more with a soil environmental-driven stabilization mechanism. The focus of this manuscript are C stabilization mechanisms which are closely related to mineral-organic interactions and how they differ across soil geochemistry. Since we can exclude recent deposition of soil material from slopes in the valleys, we infer that valley SOC stocks are more driven by soil hydrological conditions than by mineral-associated C stabilization which are dominant at non-valley positions (see manuscript L350-353). In addition, the content of pedogenic oxides in valleys is similar or even less compared to non-valley positions. Therefore, higher valley SOC stocks cannot be explained by more abundant stabilization partners or thereby enhanced microaggregation. Thus, we interpret higher SOC stocks in the valley compared to non-valley positions as a result of reduced decomposition rates caused by variations in e.g. soil moisture content (see supplementary discussion).

However, if the editors share the same opinion as the reviewer, we happily incorporate the supplements to the manuscript.

*REVIEWER#3 COMMENT 8: "Lines 350-353. The inference that "Even though valley positions are of the same geochemistry as the non-valley positions, geochemical soil properties in valleys were significantly different than at non valley positions, as fluvial activity and sedimentation unrelated to hillslope processes were dominant", seems quite speculative to explain higher C stocks in valleys relative to non-valley positions. In my opinion, a predominant effect of "fluvial and sedimentation" rather than "hillslope processes" would make sense only if the geochemistry in the valleys were significantly different from that observed in non-valley positions."*

**Our response:** Thanks for this comment. I would kindly refer to the above response to reviewer#3 comment 1 and 7 as it deals with a similar topic. We admit that our conclusion is based on qualitative field observations and descriptive data of SOC stocks and oxide content and not on statistical tests due to the small sample size for valley positions and missing soil hydrological data (e.g volumetric soil moisture content, electrical conductivity, soil temperature). But we can observe very low pedogenic oxides combined with high SOC stocks in the valley positions of the mixed sedimentary region. Here, valleys are characterised by a high groundwater table. It is most likely that the majority of pedogenic oxides are reduced and washed away by fluvial dynamics as shown by the gleyic properties which we noted during our profile description work. Furthermore, valley bottoms might be affected by material redistribution in the entire catchment during flood events overprinting the mineralogical effect of the

parent material on SOC dynamics as indicated by changes in gravel content and texture. Thus, soil moisture conditions at the valley bottoms and temporarily high groundwater levels may be restricting microbial activity. Fluvial and hydraulic conditions then lead, indirectly, to higher SOC stocks there than at the non-valley positions. This conclusion is rather hypothetical but provides the most likeable and easiest explanation with the data on hand.

*REVIEWER#3 COMMENT 9: "Lines 410-411. In the sentence "Note that while SOCbulk decreased strongly with depth in the mafic and felsic region, only a weak decrease of SOCbulk with depth was observed in the mixed sedimentary region (Fig. 4)", can we infer that SOC buildup followed the accumulation of sediments over time to a greater extent than C inputs from the local vegetation? How does the 14C depth-trend compare to that observed in valley positions as shown in Fig. S2?"*

**Our response:** Thanks for this interesting question. We respectfully disagree with the inference that SOC stock buildup followed the accumulation of C bearing sediments and outpaced C input of local vegetation since we are talking about very different time scales. The deposition of organic material and its diagenetic transformation (e.g. kerogen transformation and coalification) happened over geological timescales (several 100 Ma years; Schlüter and Trauth, 2006) long before pedogenesis and vegetation evolution of the recent soil system started, whereas the turnover of recent C input happens on a decadal to millennial time scale (Trumbore, 2000; Trumbore, 2009). The authors would argue that during soil formation the topsoil SOC stocks are dominated by faster cycling plant-derived C pools whereas the subsoil SOC stocks are more influenced by FOC released from the sedimentary rock which cycles much slower when entering microbial-mediated SOC dynamics (probably due to the high energy demand for breaking down FOC). Interestingly, we observe comparable $\Delta^{14}C$ depth trends both in non-valley and valley positions which may indicate that FOC is not varying in its persistence with topography and changing environmental conditions.

*Used Literature:*

Schlüter, T., and Trauth, M. H.: Geological atlas of Africa: with notes on stratigraphy, tectonics, economic geology, geohazards and geosites of each country, Springer, Berlin, New York, 272 pp., 2006.

Trumbore, S.: Age of soil organic matter and soil respiration: radiocarbon constraints on belowground C dynamics, Ecol. Appl., 10, 399-411, https://doi.org/10.1890/1051-0761(2000)010[0399:AOSOMA]2.0.CO;2, 2000.

Trumbore, S.: Radiocarbon and soil carbon dynamics, Annu. Rev. Earth Planet. Sci., 37, 47-66, 10.1146/annurev.earth.36.031207.124300, 2009.

*REVIEWER#3 COMMENT 10: "Lines 471-472. Based on the observation that "Soil depth and rPC4nv explained 73 % of variability (R2 ) in SOCbulk (p < 0.01). Soil depth contributed 82 % to the explanatory power of the model", how (in)sensitive tropical C pools may be to changes in climate or land use?"*

**Our response:** This is indeed a very relevant question. In general, tropical soils are very sensitive to land use change and SOC stocks are decreasing after conversion from forest to cropland as a result of reduced C input and tillage-induced destabilization of mineral-protected C (Guillaume et al., 2015; Jackson et al., 2017). At the same time, tropical soils strongly depend on this stabilization mechanism since C input is prone to fast decomposition (Zech et al., 1997). Our forest study sites are located in a highly undulated landscape under comparably similar climate across the three study regions. In regard to the high rainfall erosivity of our study region (Panagos et al., 2017), soils are especially vulnerable to erosion if deforestation continues to progress. We conclude that the studied soils are sensitive to land

use change especially since most SOC is stored in topsoils, which are affected directly by disturbances. However, this is the subject of ongoing work.

*Used Literature:*

Guillaume, T., Damris, M., and Kuzyakov, Y.: Losses of soil carbon by converting tropical forest to plantations: erosion and decomposition estimated by δ13C, Global Change Biol., 21, 3548-3560, 10.1111/gcb.12907, 2015.

Jackson, R. B., Lajtha, K., Crow, S. E., Hugelius, G., Kramer, M. G., and Pineiro, G.: The ecology of soil carbon: pools, vulnerabilities, and biotic and abiotic controls, Annu. Rev. Ecol., Evol. Syst., 48, 419-445, https://doi.org/10.1146/annurev-ecolsys-112414-054234, 2017.

Panagos, P., Borrelli, P., Meusburger, K., Yu, B., Klik, A., Lim, K. J., Yang, J. E., Ni, J., Miao, C., Chattopadhyay, N., Sadeghi, S. H., Hazbavi, Z., Zabihi, M., Larionov, G. A., Krasnov, S. F., Gorobets, A. V., Levi, Y., Erpul, G., Birkel, C., Hoyos, N., Naipal, V., Oliveira, P. T. S., Bonilla, C. A., Meddi, M., Nel, W.,Dashti, H. A., Boni, M., Diodato, N., van Oost, K., Nearing, M., and Ballabio, C.: Global rainfall erosivity assessment based on high-temporal resolution rainfall record, Sci. Rep., 7, 1-12, 10.1038/s41598-017-04282-8, 2017.

Zech, W., Senesi, N., Guggenberger, G., Kaiser, K., Lehmann, J., Miano, T. M., Miltner, A., and Schroth, G.: Factors controlling humification and mineralization of soil organic matter in the tropics, Geoderma, 79, 117-161, 10.1016/S0016-7061(97)00040-2, 1997.

*REVIEWER#3 COMMENT 11: "It looks quite amazing that when the effect of depth is controlled, the explanatory power of the other variables included in the model does not increase substantially (except silt content). How does this trend compare to temperate ecosystems? What can be inferred about the relationship between pedogenesis and soil C accumulation in the tropics? What is the mineralogy of the silt fraction? Given the data shown in Table 3 and Figure 5, such information is very important for this research and would facilitate the discussion (lines 570-578)."*

**Our response:** Thanks for your comment. It could mean that the relationship between the target and already included explanatory variables in the regression model were not masked by soil depth to such an extent compared to the excluded variables. In general, we observe that our explanatory variables gain or retain their prediction power in the partial correlation analysis pointing to their importance in explaining the target variables at all soil depths as stated in the manuscript (L543-544). This is comparable to a study comparing mineral-related C stabilization in top- and subsoils of temperate and tropical soils (Jagadamma et al., 2014). It shows that the mineral matrix is equally reactive and relevant for C stabilization at all soil depths like in our study. Similarly, Angst et al. (2018) demonstrated that in temperate soils clay content is an important predictor for SOC stock independent of soil depth.

As discussed at the end of section "4.3 Interpreting soil controls for predicting SOC dynamics", mineral-related stabilization processes are important in both tropical and temperate soils. However they are more important in the tropics for sustaining SOC stocks since C input is less compared to the temperate zone due to higher microbial decomposition rates (Zech et al., 1997). We infer that the weathering stage of tropical soils and therefore the amount of reactive mineral surfaces is highly important as it determines the size of the mineral-related C sink. Less weathered tropical soils (due to geological or anthropogenic disturbance) will have a low C sequestration potential since stabilization partners are rare and the accumulation of unprotected C is counteracted by high decomposition rates and vice-versa.

Until now, we don´t have XRD-data available on the mineralogy of the silt fraction. In general, the silt fraction of tropical soils consist of primary minerals like quartz, plagioclase, orthoclase, mica, illite and secondary minerals like gibbsite, kaolinite and pedogenic oxides in which the specific composition depends on the parent material (Martinez and Souza, 2020; Soares et al., 2005). We agree with the

reviewer that additional mineralogical data on the silt and also clay fraction would facilitate the discussion. Regarding such labour-intensive work, we would declare this task as future work.

*Used Literature:*

Angst, G., Messinger, J., Greiner, M., Häusler, W., Hertel, D., Kirfel, K., Kögel-Knabner, I., Leuschner, C., Rethemeyer, J., and Mueller, C. W.: Soil organic carbon stocks in topsoil and subsoil controlled by parent material, carbon input in the rhizosphere, and microbial-derived compounds, Soil Biol. Biochem, 122, 19–30, https://doi.org/10.1016/j.soilbio.2018.03.026, 2018.

Jagadamma, S., Mayes, M. A., Zinn, Y. L., Gísladóttir, G., and Russell, A. E.: Sorption of organic carbon compounds to the fine fraction of surface and subsurface soils, Geoderma, 213, 79–86, https://doi.org/10.1016/j.geoderma.2013.07.030, 2014.

Martinez, P., and Souza, I. F.: Genesis of pseudo-sand structure in Oxisols from Brazil – a review, Geoderma Regional, 22, https://doi.org/10.1016/j.geodrs.2020.e00292, 2020.

Soares, M. J., Alleoni, L. R. F., Vidal-Torrado, P., and Cooper, M.: Mineralogy and ion exchange properties of the particle size fractions of some Brazilian soils in tropical humid areas, Geoderma, 125, 355-367, 10.1016/j.geoderma.2004.09.008, 2005.

Zech, W., Senesi, N., Guggenberger, G., Kaiser, K., Lehmann, J., Miano, T. M., Miltner, A., and Schroth, G.: Factors controlling humification and mineralization of soil organic matter in the tropics, Geoderma, 79, 117-161, 10.1016/S0016-7061(97)00040-2, 1997.

*REVIEWER#3 COMMENT 12: "To what extent the inference that "In contrast to our initial hypothesis that topography affects C stabilization in tropical forest soils through lateral material movements, we found no indication of this in our analysis (Supplementary results and short discussion therein)" can be reconciled with the observation of higher SOC stocks in valley positions, despite exhibiting similar geochemistry to non-valley positions?"*

**Our response:** Thanks for this comment. To reduce redundancy in the responses, we would kindly refer to the discussion of reviewer#3 comments 1, 7 and 8. The simplest even though hypothetical explanation for higher SOC stocks in the valley positions is the environmental restriction of microbial activity independent of geochemical soil properties or lateral material fluxes along the catenae. Changes in gravel content with soil depth and sedimentary layering combined with gleyic properties observed during soil profile description indicate that valleys are more driven by fluvial activity which overprints the geochemical effect on SOC dynamics.

*REVIEWER#3 COMMENT 13: "Lines 410-412. "Differences in Δ14C were best explained with soil depth and the presence of FOC, which appears to be decomposable by microbial communities under more fertile, topsoil conditions." There is an apparent redundancy here since Δ14C would co-variate with FOC and factors limiting microbial respiration at depth should be more important than soil fertility."*

**Our response:** Thanks for pointing this out. The authors agree with the statement that factors limiting microbial respiration with depth are also important in explaining $\Delta^{14}C$ patterns. We would expand the section in following way:

"Differences in $\Delta^{14}C$ were best explained by soil depth as a proxy for factors limiting microbial respiration, which are more pronounced in sub- than in topsoils. The presence or absence of FOC explains $\Delta^{14}C$ patterns when comparing across regions."

We hope we have addressed all concerns and look forward to hearing from you.

 Best regards,

The authors